# The molecular pH-response mechanism of the plant light-stress sensor PsbS

Maithili Krishnan-Schmieden [1,3], Patrick E. Konold [2,3], John T. M. Kennis [2✉] & Anjali Pandit [1✉]

Plants need to protect themselves from excess light, which causes photo-oxidative damage and lowers the efficiency of photosynthesis. Photosystem II subunit S (PsbS) is a pH sensor protein that plays a crucial role in plant photoprotection by detecting thylakoid lumen acidification in excess light conditions via two lumen-faced glutamates. However, how PsbS is activated under low-pH conditions is unknown. To reveal the molecular response of PsbS to low pH, here we perform an NMR, FTIR and 2DIR spectroscopic analysis of *Physcomitrella patens* PsbS and of the E176Q mutant in which an active glutamate has been replaced. The PsbS response mechanism at low pH involves the concerted action of repositioning of a short amphipathic helix containing E176 facing the lumen and folding of the luminal loop fragment adjacent to E71 to a $3_{10}$-helix, providing clear evidence of a conformational pH switch. We propose that this concerted mechanism is a shared motif of proteins of the light-harvesting family that may control thylakoid inter-protein interactions driving photoregulatory responses.

[1] Dept. of Solid-State NMR, Leiden Inst. of Chemistry, Leiden University, Leiden, The Netherlands. [2] Department of Physics and Astronomy, Faculty of Sciences, Vrije Universiteit, Amsterdam, The Netherlands. [3] These authors contributed equally: Maithili Krishnan-Schmieden, Patrick E. Konold. ✉email: j.t.m. kennis@vu.nl; a.pandit@chem.leidenuniv.nl

Photosynthetic efficiency is tightly coupled with plant fitness under fluctuating environmental conditions, which is a prerequisite for survival in a changing global climate. Key to vitality is a continuous balance between light excitation and substrate availability that prevents the formation of lethal reactive oxygen species. Excess light conditions cause acidification of the chloroplast luminal compartments. This provokes a kinetic switch of the photosynthetic antenna into a dimmer state, where excitations are rapidly quenched by a process called non-photochemical quenching (NPQ)[1,2]. Whereas the photo-protective feedback response is beneficial for plant fitness, a large part of the absorbed solar energy is dissipated during the photoprotective state and its transient recovery[3,4].

The heart of the photoprotective mechanism lies in the molecular response to a transmembrane pH gradient (ΔpH). The protein PsbS was identified as gene product required for NPQ activation in plants[5]. PsbS is a member of the greater LHC superfamily, but in contrast to all other proteins in this family, it does not specifically bind any chlorophyll or xanthophyll pigments[6,7]. PsbS acts as a molecular pH sensor that senses thylakoid lumen acidification and transfers the signal to the antenna, enabling a switch into a photoprotective state via antenna-photosystem rearrangements that involve changes in protein–protein or pigment–protein molecular interactions[8–14]. The light-stress signaling function of PsbS and its impact on photoprotective feedback activation offers prospective for targeted crop engineering. A recent study demonstrated that a ~15% crop yield increase could be achieved in engineered tobacco plants with increased levels of PsbS, violaxanthin de-epoxidase (VDE), and zeaxanthin epoxidase (ZEP) that had faster photoprotective response kinetics[15]. Moreover, raising PsbS levels was shown to increase water use efficiency of tobacco plants[16] and canopy radiation use efficiency in rice plants[17].

PsbS activation involves two lumen-faced glutamate (Glu) residues and is associated with reversible monomerization of PsbS dimers[13,18–20]. The low-pH X-ray structure shows a PsbS dimer[20] and in detergent solutions, PsbS monomers and dimers are formed[21]. This suggests that in the absence of other partner proteins, PsbS can interact with itself.

PsbS consists of four transmembrane (TM) helices and two short amphipathic helical stretches (H1 and H2) at the luminal side and the structure displays an overall pseudo-two-fold symmetry (see Fig. 1a). The luminal loop adjacent to H1 contains a $3_{10}$ short helix domain, which was named H3 in the recent work of Liguori et al.[19]. On the basis of the high sequence identity between the two halves of the protein together with the NPQ responses of site-directed Glu mutants in vivo, the two active Glu, E69 and E173 in spinach PsbS, were proposed to be equivalent in their response to pH: the mutants E69Q and E173Q both exhibited significantly reduced NPQ responses, while the double mutant E69Q/E173Q did not show any appreciable NPQ response[14]. An in-silico study of Liguori et al. predicted that the H3 domain adjacent to E69 switches to a luminal loop at neutral pH, and proposed that this could modulate inter-protein electrostatic interactions as function of pH[19]. According to the molecular dynamics (MD) simulation, the lumen-faced Glu have unusually high pKa values, providing an explanation for the pH responsiveness to thylakoid lumen acidification of this protein domain.

Despite the important roles identified of PsbS for the fitness of plants and optimizing biomass production, the molecular change underlying the predicted pH switching mechanism has not yet been clarified. PsbS has not been crystallized under neutral pH conditions[20], which is likely due to increased dynamics compared to its low-pH state[20,21]. In contrast to other light-harvesting proteins, its lack of pigments does not provide PsbS with intrinsic fluorescent probes to report on its function. Moreover, the low occurrence of PsbS in native membranes and its poor stability in commonly used detergents or membrane protein solubilization complicates its purification in high yields for structural studies. In this work, we overcome the experimental challenges for studying the PsbS molecular mechanism and uncover its pH-dependent response mechanism. We recently introduced strategies for large-scale recombinant production of PsbS and its detergent-based refolding into native-like protein complexes[21]. In this work, we utilize those methods together with solid-state NMR and vibrational spectroscopies to evaluate the pH-dependent structure and dynamics of wild-type (WT) PsbS from *Physcomitrella patens* and that of a site-directed glutamate mutant, known to have impaired activity in vivo. Our results reveal key conformational changes that underlie the pH-dependent response mechanism of PsbS.

## Results and discussion

Three *P. patens* PsbS mutants were constructed, in which the active Glu were mutated to Gln: E71Q, E176Q (the equivalents of E69Q and E173Q in spinach PsbS) and the double mutant E71Q/E176Q, further denoted as M1 (E71Q), M2 (E176Q), and M1/M2 (E71Q/E176Q). The two active Glu residues E71 and E176 are further denoted as Glu-1 and Glu-2. In plants, single mutations of either of the two conserved Glu resulted in reduced NPQ activity, while double mutants exhibit almost complete abolishment of NPQ[14]. The WT PsbS and M1, M2, and M1/M2 mutants were refolded in *n*-dodecylphosphocholine (FC-12) detergent micelles as described earlier for the wild type[21] and protein-detergent solutions were equilibrated at pH 7.5 or pH 5.0 in 100 mM sodium phosphate buffer (pH 7.5) or 100 mM sodium acetate buffer (pH 5.0). The two different buffers were chosen for optimal buffering capacity at the respective pH conditions. We selected pH 5.0 as the low pH condition to allow a direct comparison with the X-ray structure, which was resolved at pH 5.0, even though this is below the physiologically relevant pH range within which PsbS seems to work in vivo[22]. Supplementary Fig. 1 presents a sodium dodecyl sulfate polyacrylamide gel electrophoresis (SDS)-page gel electrophoresis analysis of the four samples under pH 7.5 and 5.0 conditions directly after protein refolding in FC-12 buffer. Dimer bands are observed, in agreement with previous studies that PsbS can form strong dimers that resist gel denaturation and boiling[21,23]. The mutants had lower dimer content than the wild type and in particular mutation of Glu-1 only (M1) resulted in almost complete absence of dimers on gel.

**Protonation states of PsbS assessed by solid-state NMR.** To determine PsbS protonation states at pH 7.5 and 5.0 conditions, we analyzed uniformly isotope-labeled $U$-$^{13}$C-$^{15}$N PsbS by solid-state Magic-Angle Spinning (MAS) NMR spectroscopy. Figure 2 presents 1D $^{13}$C MAS-NMR spectra of PsbS at neutral and low pH. Direct $^{13}$C polarization MAS NMR was applied because the dimensions of PsbS in detergent micelles are such that rotational correlation dynamics averages dipolar nuclear couplings, reducing the efficiency of cross-polarization-based MAS NMR, while the sizes are too large for detection of rigid protein sites by solution NMR. At pH 7.5, two clear peaks at 181.7 ppm and 178.0 ppm are observed of deprotonated Glu and Asp carboxylic-acid side chains (Fig. 2a). At pH 5.0 (Fig. 2b), these two peaks fully disappear while gain of signal is observed of shoulders at 179.3 and 177.1 ppm. It is well known that protonation of Glu and Asp induces an upfield shift of the carboxyl $^{13}$C NMR resonances[24,25]. We therefore can interpret the signal change as an indication that most of the titratable Asp and Glu residues, 19 in total for *P. patens* PsbS, are protonated at pH 5.0. The shifted Asp and Glu signals are partly hidden under the protein carbonyl peak

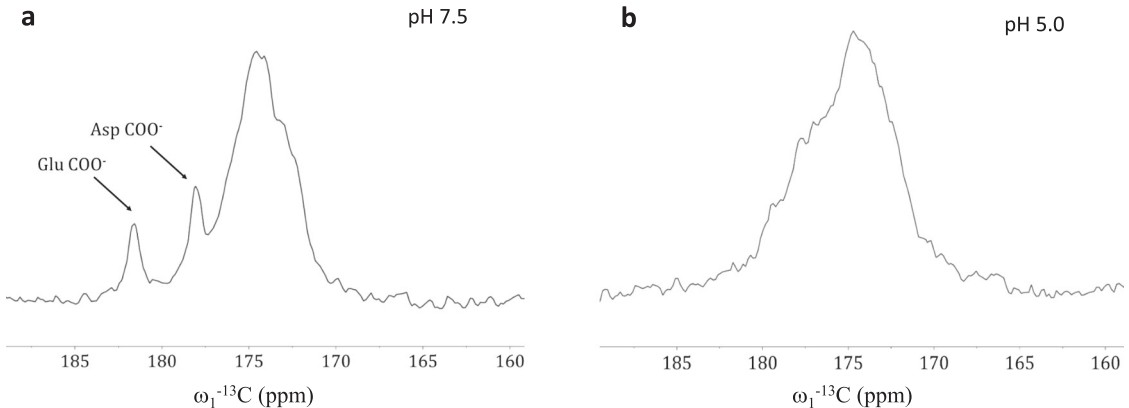

**Fig. 1 Structure and sequence of PsbS.** PsbS structure, sequence, location of active Glu-1 and Glu-2 and intradimer hydrogen bond interactions. **a** Structure of spinach PsbS (*PDB-ID 4RI2*[20]), highlighting the active Glu (E69 and E173 in yellow). Part of the stromal loop between TM2 and TM3 is not resolved in the X-ray structure. **b** Amino acid sequences of spinach (*S. oleracea*) and *P. patens* PsbS, with the two active Glu highlighted in red. **c** X-ray structure of the PsbS dimer with the active Glu-1 (E69, in yellow) and Glu-2 (E173, in orange). **d** Detail of dimerization interface viewed from the luminal side. Helix H2 containing Glu-2 (E173) connects with the luminal loop containing Glu-1 (E69) of the adjacent monomer, via hydrogen bonds from the Y75 backbone amide to the E173 side chain and the I74 backbone amide to the E173 backbone carbonyl. **e** Schematic picture of the inter-monomer stabilizing interactions at the luminal side, involving Glu-2 (E173) in helix H2 and the 3$_{10}$ helix/luminal loop containing Glu-1 (E69).

**Fig. 2 NMR spectroscopy of PsbS.** Glu and Asp protonation states at pH 7.5 and 5.0 probed by NMR spectroscopy. **a** 1D $^{13}$C-DP MAS NMR spectra of PsbS at pH 7.5, **b** 1D $^{13}$C-DP MAS NMR spectra of PsbS at pH 5.0. The spectra contain the sum of 1024 accumulated scans and a line broadening function of 50 Hz was applied for processing.

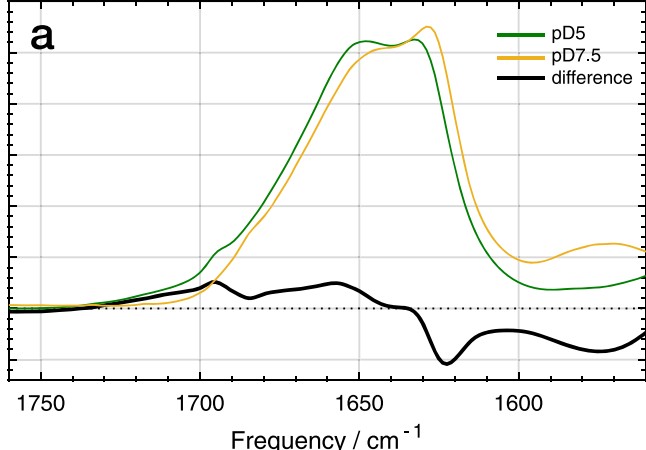

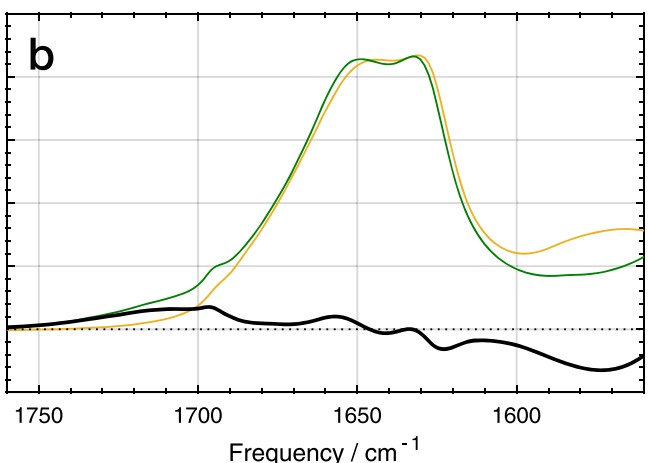

**Fig. 3 FTIR spectroscopy of PsbS.** FTIR spectra of WT PsbS and the M2 (E176Q) mutant. **a** FTIR spectra of WT PsbS at pD 7.5 (orange), pD 5.0 (green), and the difference spectra of pD 5.0 minus 7.5 (black). **b** FTIR spectra of the M2 (E176Q) mutant at pD 7.5 (orange), pD 5.0 (green), and the difference spectra of pD 5.0 minus 7.5 (black). All spectra were taken in $D_2O$ buffer. 100–150 scans were taken in one acquisition of each pD and repeated 10 times. 6 and 3 protein samples were investigated for WT and M2 mutant, respectively, of which one is shown for each pD.

centered at 174.6 ppm. Below, we will see that the FT-IR results unambiguously show protonation of all protonable residues at low pH, consistent with the NMR results. MD simulations of PsbS over a whole pH range predicted that only the luminal Glu residues protonate in the pH range 5–7 and that the Glu residues located in the stromal loops at the aqueous phase have lower pKa values, resembling those of Glu in a water environment[19], at odds with the present results.

**FTIR spectroscopy of PsbS at neutral and low pH.** To explore the possibility of a secondary-structure response of PsbS to a change in pH, we performed FTIR and 2DIR spectroscopy in the Amide I region. The Amide I bands arise mainly from $C=O$ oscillators in the protein backbone, and their frequencies are sensitive to particular secondary structure elements (α-helix, β-sheet, and loops/turns) and their microenvironment, e.g., the degree of solvent exposure[26]. The spectra were collected in $D_2O$ detergent buffer solution rather than $H_2O$ to avoid the O-H bend absorption which overlaps with Amide I. For this reason, the description of the FTIR results will be discussed in terms of pD rather than pH. Figure 3 shows the FTIR spectra of WT PsbS and the M2 mutant at pD 7.5 (orange lines) and pD 5.0

(green lines) conditions, together with the difference spectrum of pD 5.0 minus 7.5 (black lines). The wild-type FTIR spectrum at pD 7.5 (Fig. 3a, orange line) shows that the Amide I band is centered at $1640\,cm^{-1}$ with a broad shoulder at frequencies up to $1690\,cm^{-1}$, indicative of predominant helical and loop contributions[26], consistent with the X-ray structure[20] and CD data[21]. Importantly, the spectrum contains a sharp band at $1630\,cm^{-1}$ that will be discussed in detail below. In addition, the pD 7.5 FTIR spectrum shows a prominent band at $1570\,cm^{-1}$ that originates from deprotonated carboxylic acid (the $COO^-$ stretch mode), representative of deprotonated Glu and Asp residues[27]. In the spectral region around $1570\,cm^{-1}$, also the Amide II mode is expected to contribute, which arises from the backbone $C=N$ stretch coupled to the amide N-H bend. However, because the protein has been dissolved in $D_2O$, the Amide II band is downshifted by $100\,cm^{-1}$ to $\sim1450\,cm^{-1}$ [28], outside of the probed spectral window.

Significant changes are observed in the FTIR spectrum for WT PsbS at lower pD. At pD 5.0 (Fig. 3a, green line), the $1570\,cm^{-1}$ band is entirely absent while a new shoulder from $\sim1700$ to $1750\,cm^{-1}$ is observed, indicative of protonated (or in this case, deuterated) carboxyls (COOH/COOD)[27]. This observation demonstrates that all, or nearly all Glu and Asp are protonated at this pH, in agreement with the NMR results (Fig. 2b).

In addition to the protonation effects, the pD 5.0 – minus - pD 7.5 difference spectrum exhibits changes in the Amide I region from 1610 to $1690\,cm^{-1}$. It shows a negative sharp signal at $1625\,cm^{-1}$ and a broad, positive band centered at $1660\,cm^{-1}$, which may be assigned a change in the Amide I band, and hence to conformational changes in the PsbS secondary structure. The integrated bleach amplitude of the $1625\,cm^{-1}$ band corresponds to 3.6% of the integrated Amide I absorption, and with a total of 221 amino acids in *P. patens* PsbS, hence involves at least 8 amino acid backbone oscillators, indicating that a conformational change occurs involving at least 8 amino acid residues. Note that because these 8 amino acids absorb at the same particular vibrational frequency at $1625\,cm^{-1}$, which is unusually low, they are likely part of a specific secondary structure element.

In contrast to the WT spectrum, in the FTIR spectrum of the M2 mutant only small changes are observed in the Amide I region comparing the two pD conditions, with a minor negative band at $1625\,cm^{-1}$, and an even smaller negative/positive feature around 1645 and $1658\,cm^{-1}$ in the difference spectrum (Fig. 3b, black line). Remarkably, the Amide I spectra of M2 under both pD conditions are similar to that of the wild type at low pD (Supplementary Fig. 2), indicating that the M2 mutant resides in a conformational state that resembles (but is not identical to) the WT state at low pH. The deprotonated carboxyl signals at $1570\,cm^{-1}$ (negative) and the protonated carboxyl signals at $1700–1750\,cm^{-1}$ (positive) are conserved, indicating that also in this mutant, lowering of the pD induces protonation of titratable Glu and Asp residues. These observations indicate that the majority of the backbone conformational changes that are observed in WT PsbS on lowering the pD from 7.5 to 5.0 are specifically triggered by protonation of Glu-2.

In the WT FTIR difference spectrum (Fig. 3a), a slight derivative-like signal around $1690\,cm^{-1}$ is observed, representing a low-amplitude modulation of the larger positive induced absorption signal caused by Glu/Asp acid protonation at low pD. We currently do not know the origin of the modulation signal.

The Amide I spectra of the M1 and M1/M2 mutants (Supplementary Fig. 3a,b, respectively) do not resemble the WT spectra at either pD conditions. In particular, the prominent band around $1630\,cm^{-1}$ that is conspicuously present in WT at pD 7.5 and to a lesser extent in WT at pD 5.0 and in the M2 mutant at both pD conditions, has a lower amplitude as compared to the

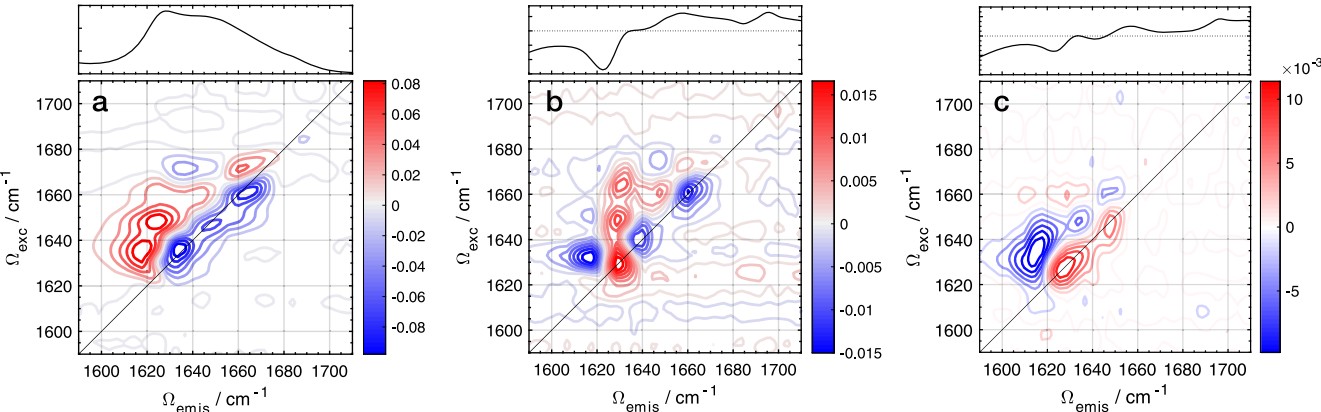

**Fig. 4 2DIR spectroscopy of PsbS.** 2DIR spectra of WT PsbS and the M2 (E176Q) mutant. **a** Equilibrium spectrum of WT PsbS in pD 7.5 buffer; upper panel: FTIR absorption spectrum of WT PsbS in pD 7.5 buffer; **b** pD 5.0 – 7.5 difference 2DIR of WT PsbS; upper panel: pD 5.0 – 7.5 difference FTIR of WT PsbS; **c** pD 5.0 – 7.5 difference 2DIR of the M2 (E176Q) mutant. upper panel: pD 5.0 – 7.5 difference FTIR of the M2 (E176Q) mutant. For the WT, 15000 scans were taken per sample at each pD and 6 protein samples were studied, of which one is shown. For the M2 mutant, 15000 scans were taken per sample at each pD and 3 protein samples were studied, of which one is shown.

band at 1655 cm$^{-1}$ in the M1 and M1/M2 mutants. Thus, parts of the M1 and M1/M2 PsbS mutants adopt a non-native fold, as will be discussed in a separate section later in the manuscript.

**2DIR spectroscopy of WT and E176Q (M2) PsbS.** The FTIR results clearly indicate specific conformational changes in the PsbS secondary structure, but they are ambiguous with respect to the type of secondary structure element (i.e., helix, sheet, or loop/coil) that undergoes a change or is newly formed. In particular, the negative feature at 1625 cm$^{-1}$ might correspond to either β-sheet or α-helix, and the positive feature at 1660 cm$^{-1}$ could correspond to a helix or a coil/loop[26]. Note that even though the PsbS X-ray structure indicates an all-helical and loop structure, it was determined only at low pH and some parts were not resolved[20], thus, a priori one cannot exclude the occurrence of β-sheet elements. To resolve any ambiguities, we applied two-dimensional IR (2DIR) spectroscopy[29], which has a larger resolving power as compared to FTIR in this regard. In the last decades, this method has emerged as a powerful means of characterizing small peptides and model proteins[30,31], but have rarely been applied for the analysis of integral membrane proteins[32,33]. The steady-state 2DIR spectrum of WT PsbS at pD 7.5 is presented in Fig. 4a, with the excitation frequency $\Omega_{exc}$ along the vertical axis and the emission frequency $\Omega_{emis}$ along the horizontal axis. The linear FTIR spectrum is shown in the upper panel, reproduced from Fig. 3a. The 2DIR spectrum exhibits spectral congestion as typically observed in several other protein systems due to the overlapping Amide I contributions of several structural elements[30,31]. Adjacent to the negative bleach/stimulated emission bands along the diagonal, reflecting depopulation of ground state modes $v = 0 \longrightarrow 1$, we observed corresponding anharmonic peaks of positive sign, corresponding to the $v = 1 \longrightarrow 2$ transition which occurs as a result of population of the $v = 1$ vibrational level and slightly red-shifted along the $\Omega_{emis}$ axis. This stems from the fact that the energy spacing between the $v = 1$ and $v = 2$ vibrational levels is smaller than that between $v = 0$ and $v = 1$ given the anharmonicity of the potential energy well[29].

We recorded equilibrium 2DIR spectra at pD 5.0 and 7.5, and determined the difference spectrum (Fig. 4b). Supplementary Fig. 4 shows the equilibrium 2DIR spectra at pD 5.0 (A) and 7.5 (B) side-by-side. The FTIR difference spectrum is shown in the upper panel. Note that the sign of the 2DIR difference pattern is inverted with respect with that of FTIR, since in the former case, the signals on the diagonal in the equilibrium spectra are negative

as explained above[29]. The 2DIR difference spectrum shows significantly less spectral congestion as several discrete peaks are observed reflecting specific conformational changes that occur as a function of pD. Beginning on the low frequency side, there is a $v = 0 \longrightarrow 1$, $v = 1 \longrightarrow 2$ positive/negative peak pair centered at ~1630 cm$^{-1}$, nearly coinciding with the negative feature identified in the difference FTIR spectrum at 1625 cm$^{-1}$. Accordingly, the sign of the 1630 cm$^{-1}$ band in the difference 2DIR map is positive, reflecting population loss with the positive lobe falling on the diagonal, which represents loss (or rather rearrangement, as shown below) of a specific secondary structure element. Two diagonal peak pairs of opposite signs with respect to that at 1630 cm$^{-1}$ are located at 1638 and 1660 cm$^{-1}$, i.e. with a negative amplitude on the diagonal, which indicates that these pairs represent population gain of two distinct secondary structure elements. The 1660 cm$^{-1}$ band has an amplitude that is somewhat (~30%) higher than that at 1638 cm$^{-1}$. As discussed below, these two bands at 1660 and 1638 cm$^{-1}$ represent a gain or rearrangement of distinct helical elements, with different degrees of conformational disorder.

The nature of the transition at 1630 cm$^{-1}$ warrants further attention, since this frequency falls on the boundary expected for α-helical and β-sheet Amide I oscillators[26]. A more precise assignment can be made by carefully inspecting the 2DIR spectrum in Fig. 4b. First, the band anharmonicity ($\Delta$) is an important parameter that reflects the extent of backbone Amide coupling and can help distinguish between structured and unstructured elements. The anharmonicity $\Delta$ of the 1630 cm$^{-1}$ transition here was determined to be 13 cm$^{-1}$ on the basis of the difference of the $v = 0 \longrightarrow 1$ and $v = 1 \longrightarrow 2$ peak maxima. Uncoupled Amide I oscillators carry $\Delta$ values of 16 cm$^{-1}$ and this value inversely scales with the degree of excitonic coupling between Amide I oscillators[34,35]. For example, an α-helix or β-sheet generally ranges from 9–14 cm$^{-1}$ suggesting this element corresponds to an ordered structure and is therefore not representative of a coil or loop. Second, we can discriminate between a β-sheet and helical response on the basis of its peak pattern. Since β-sheets typically exhibit a characteristic "Z-shaped" peak pattern in the overall 2DIR spectrum, which is not observed here, we conclude that the 1630 cm$^{-1}$ feature corresponds to a helical element[36]. This is in agreement with the X-ray structure and CD data of PsbS which indicate no extensive β-sheet content[20,21]. A similar reasoning may be applied to the two band pairs at 1638 and 1660 cm$^{-1}$, which

show similar band anharmonicity and peak patterns to that at 1630 cm$^{-1}$ indicating distinct helical elements. Figure S5 shows a diagonal slice of the 2DIR difference spectrum, showing the amplitudes of the bands at 1630, 1638, and 1660 cm$^{-1}$.

Comparing the 2DIR difference spectra with the FTIR difference spectra, the positive 1630 cm$^{-1}$ diagonal band in 2DIR clearly corresponds with the negative 1625 cm$^{-1}$ band in FTIR. Likewise, the negative diagonal band at 1660 cm$^{-1}$ in 2DIR corresponds to the positive 1660 cm$^{-1}$ band in FTIR. The negative diagonal 1638 cm$^{-1}$ band in 2DIR is not clearly resolved in the FTIR spectrum, yet the latter shows a positive-going amplitude shoulder at 1638 cm$^{-1}$ that is apparently super-imposed on the 1625 cm$^{-1}$ bleach. This explains the 5 cm$^{-1}$ difference in the respective band amplitude maxima in 2DIR (1630 cm$^{-1}$) and FTIR (1625 cm$^{-1}$), which likely results from compensation by positive signals at 1638 cm$^{-1}$ at the high-frequency side in the FTIR difference spectrum. The asymmetric line shape of the 1625 cm$^{-1}$ bleach (Fig. 3a) is consistent with this idea. We conclude that the FTIR and 2DIR results are mutually consistent.

Figure 4c shows the pD 5.0 minus 7.5 difference 2DIR spectrum of the M2 mutant. Supplementary Fig. 4 shows the equilibrium 2DIR spectra at pD 5.0 (C) and 7.5 (D) side-by-side. In the difference spectrum, we observe a small positive band at 1625 cm$^{-1}$ on the diagonal with a corresponding negative anharmonic band, consistent with the small remaining negative signal in FTIR (Fig. 3b). The prominent negative diagonal bands at 1638 and 1660 cm$^{-1}$ observed in the wild-type (Fig. 4b) are entirely absent, consistent with the FTIR results that pH-dependent conformational changes are largely suppressed in the M2 mutant.

An additional useful means to gain molecular insight from 2DIR spectroscopy lies in the slope of the nodal line (NLS) between the $v = 0 \longrightarrow 1$ and $v = 1 \longrightarrow 2$ transitions. This quantity is proportional to the vibrational energy gap frequency–frequency correlation function (FFCF) and reports on the degree of vibrational inhomogeneity of the system, i.e., the spread of vibrational frequencies associated with a certain oscillator[18,37,38]. We calculated the NLS of each diagonal band pair in WT PsbS by applying a linear fit to the zero crossing (nodal line) within the FWHM of each peak. The band pair at 1630 cm$^{-1}$ has a NLS nearly parallel to the $\Omega_{exc}$ axis, as illustrated in Supplementary Fig. 6, which indicates a highly homogeneous structure. In contrast, the band at 1638 cm$^{-1}$ has a NLS nearly parallel to the diagonal, representing a case with large inhomogeneity. The 1660 cm$^{-1}$ band pair demonstrates a NLS between the former examples, which can be interpreted as a case with intermediate inhomogeneity.

Our interpretation of the pD 5.0 minus 7.5 difference 2DIR results, considering their frequency, anharmonicity, inhomogeneity, and sign, is as follows:

i.   the 1630 cm$^{-1}$ diagonal peak reflects loss of a single well-ordered helical element in a homogeneous environment and corresponds to the negative signal observed in the difference FTIR spectra. These observations together with its low frequency of 1630 cm$^{-1}$ indicate that it has a high degree of exposure to the aqueous solvent[39,40].

ii.  the 1638 cm$^{-1}$ diagonal peak represents gain of a helical element. This helical element exhibits a significant degree of spectral inhomogeneity reflecting high conformational disorder.

iii. the 1660 cm$^{-1}$ diagonal peak also represents gain of a helical element. This helical element exhibits an intermediate degree of spectral inhomogeneity.

Given that the combined amplitudes of the 1638 and 1660 cm$^{-1}$ diagonal bands (gained signal) are higher than that of the corresponding band at 1630 cm$^{-1}$ (signal loss), we conclude that the helical content slightly increases at the low pD condition. Therefore, a single well-ordered helical element undergoes a change in frequency from 1630 cm$^{-1}$ at high pD to either 1660 or 1638 cm$^{-1}$ at low pD, and another helical element appears. No significant unfolding takes place, which would result in lowering of the 2DIR amplitude[41].

In addition to the diagonal peaks discussed above, the wild-type 2DIR difference map features a prominent off-diagonal peak at [1630,1665], which implies that a cross peak that exists in the equilibrium spectrum at neutral pH disappears in the equilibrium spectrum at low pH. At this point, it is difficult to pinpoint the precise origin of such cross peak, but we can generally attribute it to a change in vibrational coupling of the respective helical elements undergoing conformational change.

**The response of PsbS to low pH: repositioning of PsbS amphipathic helix H2 and refolding of $3_{10}$ helix H3.** We will now discuss the origin of the Amide I difference signals observed in FTIR and 2DIR spectroscopy. In FTIR, we observe a distinct negative/positive Amide I signal at 1625 (-) / 1660 (+) cm$^{-1}$, and a positive-going shoulder at 1638 cm$^{-1}$ in WT PsbS (Fig. 3a). This signal is also reflected and refined in 2DIR spectroscopy, where loss of a helical signal at 1630 cm$^{-1}$ is accompanied by gain of two distinct helical signals at 1638 and 1660 cm$^{-1}$ (Fig. 4b). As mentioned above, because of its low frequency, the helical signal at 1630 cm$^{-1}$ should originate from a helix element that is water exposed. This limits its assignment to the luminal amphipathic helices of PsbS (H1 and H2) as the transmembrane helices reside in a hydrophobic membrane environment. Strikingly, mutation of Glu-2 largely abolishes the structural Amide I response of PsbS to low pH (Figs. 3b and 4c), indicating that the helical signal at 1630 cm$^{-1}$ originates from the lumen-facing amphipathic helix H2 containing Glu-2 (Fig. 1a). This assignment is consistent with the FTIR amplitude of the difference signal in WT PsbS, corresponding to at least 8 amino acids, consistent with the length of the Glu-2-containing H2 helix fragment in the X-ray structure with 9 amino acids[20]. We note that H1 is much shorter and comprises 5 amino acids and therefore confidently assign the 1625 cm$^{-1}$ FTIR signal and the corresponding 1630 cm$^{-1}$ 2DIR signal in WT PsbS to a change in luminal amphipathic helix H2. Because Glu-2 is negatively charged at neutral pH, it is expected that at pH 7.5, the polar side of the amphipathic helix H2 will be stabilized in the aqueous environment of the lumen, consistent with the low frequency (1630 cm$^{-1}$) of the helical element detected at that pH (pD).

The work of Fan et al. gives evidence that at low pH, where the charge is neutralized, Glu-2 is in a hydrophobic phase. Low-pH crystal structures were obtained from crystals that were soaked with N,N'-Dicyclohexylcarbodiimide (DCCD)[20], a hydrophobic compound that binds to carboxyl groups of protonated amino acids that are shielded from aqueous environments. The DCCD was used to detect protonation sites of PsbS in the crystal structure. The DCCD-soaked structure shows that DCCD binds to Glu-2 (E173), indicating that in the low-pH conformation, this Glu residue resides in a hydrophobic environment[20]. In our results at pD 5.0, two populations of helical elements are detected with frequencies of 1660 and 1638 cm$^{-1}$. The helical signal at 1660 cm$^{-1}$ clearly indicates a hydrophobic environment. Considering the magnitude of the signal at 1660 cm$^{-1}$ and the fact that it is also largely abolished in the difference spectrum of M2, we determine that this signal appearing at low pD is associated with H2.

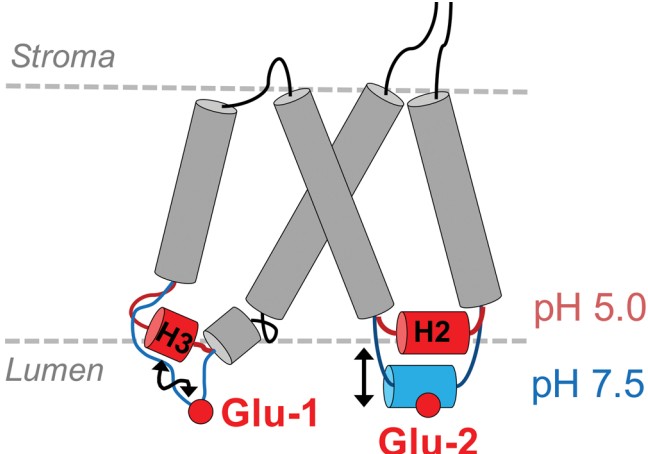

**Fig. 5 Illustration of the pH switch of PsbS.** Schematic illustration of the pH-dependent movement of helix H2 and folding of helix H3 in WT PsbS at low pH. Protonation of Glu-2 results in repositioning of amphipathic helix H2 from the aqueous phase to the membrane phase, while protonation of Glu-1 causes the H3 element to switch from a loop segment into a $3_{10}$ helix.

We therefore propose that protonation of Glu-2 at low pH causes the H2 helix to reposition from the aqueous phase into the membrane phase, as schematically illustrated in Fig. 5. This model is consistent with the M2 mutant FTIR results: there, the charge at the Glu-2 site is neutralized by the Glu to Gln mutation, which predicts that M2 PsbS will already adopt a low-pH conformation at neutral pH. This is exactly what is observed, since the Amide I FTIR spectrum of M2 at both pH conditions resembles the low-pH spectrum of the WT (Supplementary Fig. 2).

The 1638 cm$^{-1}$ helical signal detected at pD 5.0 shows that another distinct helical element is formed at low pD. Recent MD simulations on PsbS indicated a key role for the small luminal amphipathic H3 helical element[19], which assumes a $3_{10}$ helical fold in the X-ray structure and is part of the luminal loop that contains the active Glu-1 (Fig. 1a). Upon increasing the pH, the MD simulations indicated that this H3 helix was destabilized and unfolded into a loop element. A $3_{10}$ helix is expected to have its frequency around 1635 cm$^{-1}$ if exposed to D$_2$O solvent[42,43]. A loop element is expected to yield a much lower signal in 2DIR than a helical element because of the 4$^{th}$ power scaling of the IR transition dipole moment[41]. Hence, the 2DIR signal is consistent, upon going from high to low pD, with a switch from a loop element (no or low 2DIR signal) to a $3_{10}$ helix (1638 cm$^{-1}$ signal) as predicted by MD, as schematically indicated in Fig. 5.

According to the 2DIR difference spectrum, helix H2 assumes a highly ordered, homogenous state at neutral pH, as judged by the NLS (Supplementary Fig. 6). At low pH, the helix H2 band at 1660 cm$^{-1}$ shows a moderate degree of inhomogeneity, from which we conclude that its repositioning into the membrane phase is accompanied by increased conformational freedom of this protein fragment. Remarkably, the helical element that absorbs at 1638 cm$^{-1}$ and attributed to formation of H3 shows a high degree of conformational freedom, which is consistent with the propensity of $3_{10}$ helices to switch between a disordered (loop) and ordered ($3_{10}$ helical) state.

We now discuss the possible nature of the small but significant Amide I signals in the FTIR and 2DIR difference spectra of the M2 mutant. In FTIR, the small residual bleach at 1625 cm$^{-1}$ (Fig. 3b) likely represents a small change in a solvent-exposed amphipathic helix given its unusually low frequency, and may correspond to a small change in either H1 or H2. In the M2 2DIR

difference spectrum (Fig. 4c), the overall signal amplitude is also significantly diminished with respect to WT (see Supplementary Fig. 5 where diagonal slices are shown for WT and M2). Here, the difference signal is dominated by the positive band at 1625 cm$^{-1}$ with accompanying anharmonic negative band, which coincides with the 1625 cm$^{-1}$ bleach observed in FTIR. Hence, the FTIR and 2DIR results are mutually consistent and indicate that in the M2 mutant, one of the amphipathic solvent-exposed helices undergoes a small motion or minor unfolding upon lowering of the pD. The FTIR difference spectrum also involves a band upshift from 1645 cm$^{-1}$ (−) to 1655 cm$^{-1}$ (+). Its very small amplitude indicates that it may involve only a single amino acid. Likewise, the 2DIR difference map shows a very small positive diagonal band at 1650 cm$^{-1}$, indicating a small loss of helical signal. We interpret the 1645–1650 cm$^{-1}$ loss signal in FTIR and 2DIR as a loss or destabilization of transmembrane helix by a single amino acid, resulting in an upshift to 1655 cm$^{-1}$ in FTIR, which may correspond to coil or a loosened helix.

**E71Q (M1) and E71Q/E176Q (M1/M2) PsbS have non-native folds.** Next, we turn to the FTIR spectra of the M1 and M1/M2 mutants. In the FTIR spectrum of M1 at neutral pD, (Supplementary Fig. 3a, orange line) the prominent band around 1630 cm$^{-1}$ that is visible for the wild type and for M2, has a lower amplitude as compared to the main band at 1655 cm$^{-1}$. We consider two possibilities to account for this observation: (i) M1 has less water-exposed amphipathic helical content than wild type, which would imply diminished H2 or H1 absorption, or (ii) M1 has a higher transmembrane helix content as compared to WT, which increases the absorption around 1655 cm$^{-1}$. We note that in the X-ray structure, TM2 does not span the entire membrane at the luminal side (see Fig. 1a) and by mutation of Glu-1, the unstructured part adjacent to TM2 that loops into the lumen has the potential to fold into TM2, especially given that the TM2 sequence is similar to that of the pseudo-symmetry-related TM4. The spectrum of the M1/M2 mutant at neutral pD (Supplementary Fig. 3b, orange line) shows a diminished 1630 cm$^{-1}$ band relative to that at 1655 cm$^{-1}$ as well, which means that its fold may be similarly altered with respect to wild type, as in the M1 mutant.

The non-native folds of M1 and M1/M2 could be due to a non-native charge distribution in the loop connecting TM1 with TM2. At neutral pH, the negative charge on E71 is eliminated by the mutation, but other titratable residues in this loop (D69, E78, and E80 for *P. patens*) will be negatively charged, which gives a non-natural charge distribution along the loop as a whole (−4 in wild type, −3 in the M1 and M1/M2 mutants). In contrast, at the Glu-2 site, there are no additional titratable residues in the loop and H2, and elimination of the Glu-2 charge gives an overall charge of this segment that resembles the charge distribution of the wild type at low pH. In addition, the segment containing Glu-2 forms a well-defined structural element (i.e., an amphipathic helix of 9 amino acids) and its conformation should be less affected by a point mutation than the loop containing Glu-1 that is only locally structured, containing a short helix fragment H1 and a short $3_{10}$ helix fragment that carries Glu-1. $3_{10}$ helices are known to be intrinsically unstable, which is also demonstrated in the MD study of Liguori et al.[19]. A point mutation at Glu-1 therefore could easily induce an unnatural structural change.

The FTIR difference spectrum of the M1 mutant shows significant changes upon lowering the pD to 5.0 (Fig S3a, black line). The amplitude of the changes is similar to that of the wild type (Fig. 3a, black line), but it qualitatively differs significantly: it exhibits a bleach at 1629 cm$^{-1}$ (1623 cm$^{-1}$ in wild type), with a negative shoulder at 1642 cm$^{-1}$ that is absent in wild type. It

features a positive signal with a maximum at 1658 cm$^{-1}$ with an overall shape that is narrower and more symmetric than for the wild type. Likewise, the M1/M2 difference spectrum (Supplementary Fig. 3b, black line) shows a negative band at 1626 cm$^{-1}$, a negative shoulder at 1645 cm$^{-1}$ and a sharp, symmetric positive signal at 1658 cm$^{-1}$. Because M1/M2 lacks both Glu-1 and Glu-2, the observed pH (pD) dependent changes for M1/M2 must originate from responses of other titratable residues. Given the fact that the fold of M1 and M1/M2 at pD 7.5 differs from that of wild type in a way that is uncertain to gauge, we cannot clearly relate the M1 and M1/M2 difference spectra with specific elements in the wild-type X-ray structure, and we refrain from interpreting these spectra in more detail.

**Comparison with earlier MD results**. The pH-dependent motion of the luminal amphipathic H2 helix and the refolding of the H3 loop fragment into a 3$_{10}$ helix are intrinsic properties of PsbS that underlie this protein's plasticity, defined by a complex potential energy landscape that dynamically responds to changes in pH. Notably, while H3 refolding was first proposed in MD simulations of Liguori et al.[19] and consistent with the FTIR and 2DIR signals, the pH-dependent repositioning of H2 was not observed in the MD simulations. This discrepancy could have several explanations. The limited time window after pH change in the simulation, which was 4.7 µs, may have precluded such observation: stimulus-induced conformational changes in proteins may take place on much longer timescales than that, i.e., from sub-milliseconds to milliseconds[44–47], and specific methods in MD are required to predict such slow dynamics[48]. Furthermore, in the current work, the H2 motion is revealed in (primarily) dimeric PsbS in detergent micelles; the MD simulations were conducted on a PsbS monomer, as taken from the dimeric X-ray structure, and equilibrated in a POPC lipid bilayer, which renders the system flat and planar. The latter difference between our experimental systems and the MD simulations may also contribute to the discrepancies in the results with respect to protonation. The NMR and FTIR data show that all titratable residues are protonated at pH 5.0, including Glu-1 and Glu-2, which are essential for the lumen acidification response in vivo. Our results indicate that also titratable residues at the stromal site have pKa's that are significantly shifted with respect to their pKa's in aqueous solvent whereas the MD study predicts that the stromal residues have very low pKa's.

**The molecular response of PsbS to low pH in relationship with its function**. We now discuss how a pH-dependent repositioning of helix H2 and refolding of 3$_{10}$ helix H3 may be coupled to the pH-sensing function of PsbS. In the spinach PsbS crystal structure, Glu-1 (E69 in the X-ray structure) is located in a loop fragment close to H3 that connects two transmembrane helices TM1 and TM2, whereas Glu-2 (E173 in the X-ray structure) is located in the amphipathic short helix H2 at the water-membrane interface facing the lumen, and connects TM3 and TM4 (Fig. 1a,c). These elements form an important part of the PsbS dimerization interface (Fig. 1d): the backbone carbonyl and side chain carboxyl of Glu-2 each can form a hydrogen bond with two backbone amides of I74 and Y75 close to Glu-1 of the adjacent monomer, as schematically shown in Fig. 1e. In *P. patens* PsbS, the latter two amino acids are L74 and T75 (note that the different nature of the side chains in *P. patens* vs. spinach will not have an immediate effect because the hydrogen bonding takes place to the backbone amides). The I74 and Y75 residues form the short 3$_{10}$ helix fragment H3 that was predicted to unfold at neutral pH, constituting a molecular response to pH. It was proposed that including this response, a network of tunable electrostatic

interactions could create a pH-sensitive docking mechanism for PsbS[19]. Here, we have shown that helix H2 exhibits pH-dependent structural dynamics in concert with H3, suggesting that their motion forms an integral part of the pH-dependent regulation pathway. In addition to the absence of motion for H2, we find no indication of pH-dependent H3 refolding in the M2 mutant according to the 2DIR results. Thus, the luminal loop element that contains Glu-1 and partly folds into H3 at low pH in WT PsbS, has its structural pH response abolished when Glu-2 is replaced by Gln. In the M2 mutant, H3 in the 3$_{10}$ conformation is able to form a native-like dimer interaction with helix H2 of its dimerization partner. We hypothesize that in the M2 mutant, H3 retains its 3$_{10}$ conformation at neutral pH because H2 of its dimerization partner remains in the hydrophobic phase and provides an energetically favorable hydrogen bonding site, even with deprotonation of Glu-1.

Considering our results and that of earlier MD simulations[19], both Glu sites have shown to undergo specific pH-dependent conformational changes that could affect the interaction of PsbS with itself or other interaction partners in a membrane. Indeed, in-vivo studies show that both M1 and M2 mutants have altered NPQ response according to Li et al[12]. We find that at both pH conditions, the PsbS M2 mutant resides in a pseudo-WT low-pH conformational state, which is considered the active state of the protein. This is in apparent contradiction with in vivo results that indicate that mutation of Glu-2 partially impairs the NPQ response[14]. The observed partial activity in vivo may result from the function of PsbS to interact with other type of proteins, like LHCII. The M2 mutant has a fixed position of H2 but could still confer a pH-dependent switching capability through the protonation state of Glu-1 in H3, and establish reversible interactions with flexible amphipatic helices of other types of proteins. In our isolated protein system, however, PsbS can only interact with itself. This means that in the M2 PsbS dimer, the interacting partner site of H3 (i.e., the amphipatic helix H2 of the other monomer) is in a fixed position, thereby locking also the conformation and positioning of H3. In addition, the reduced NPQ activity of Glu-2 mutants in plants suggests that not only the activation of PsbS, but its ability to switch on and off is crucial for its function. Indeed, PsbS is not only involved in switching on NPQ, but also essential for rapid deactivation and recovery of this process in fluctuating light conditions[15].

In thylakoid membranes, the water-membrane interface will be determined by the compositions of native lipids in the annular shell around the protein: the non-bilayer lipid monogalactosyl diacylglycerol (MGDG), digalactosyl diacyl glycerol (DGDG), sulfoquinovosyl diacylglycerol (SQDG), and phosphatidyl glycerol (PG). The microenvironment of PsbS will change if it associates with other proteins in the membrane. The exact positioning of protein fragments at the water–lipid interface and the pKa's of titratable residues therefore may differ from experimental or in-silico results on isolated protein.

Based on our experimental results, we may extend earlier models describing the PsbS function in vivo interacting with key constituents of the thylakoid membrane[19,49,50]. It has been established that in vivo, PsbS interacts with LHCs under high light conditions[8,9,23,51]. PsbS is a member of the LHC super-family, where proteins function as pH switches (namely PsbS and LHCSR) and/or photoprotective switches that can alternate between fluorescent and excitation-quenched states (namely LHCII and LHCSR). LHCs share a structure in which transmembrane helices are connected via amphipathic short helices at the luminal site[5,52–54]. As we mentioned, in isolation PsbS self-interacts to form dimers as described above. For interactions of PsbS with LHCs, the corresponding symmetry-equivalent luminal amphipathic helices of the latter proteins

would provide a putative dimerization site. From our experimental results, we predict that (i) these LHC amphipathic helices need to be located in the membrane phase for proper positioning with respect to the PsbS helix H3 and (ii) these LHC amphipathic helices need to respond to pH (i.e., undergo a motion similar to that of H2 in PsbS) in order to tune the strength of PsbS – LHC interactions. Otherwise, H3 may remain constitutively folded as is likely the case for the PsbS M2 mutant, and the interaction is not switched on and off with varying the pH. Strikingly, the symmetry-related luminal amphipathic helices of LHC contain several titratable residues (Asp, Glu)[5,52–54] which may enable such functionality[50]. Thus, responsiveness of the amphipathic helices to changes in pH, hydrophobicity, or other alterations in the physico-chemical environment could be a common motif that enables LHCs to operate as molecular controls for regulating photosynthetic light harvesting.

Thus far, our understanding of PsbS action has mostly derived from the thylakoid membrane level; an important goal is to understand the precise mechanism by which PsbS causes NPQ on the molecular level through protonation events that trigger altered interactions with membrane partners, and how these dynamic interactions ultimately elicit the systems-level response that NPQ in essence represents. This work constitutes a first experimental characterization of the molecular pH response of PsbS, a starting point that allows to build a true molecular mechanistic model of PsbS activation.

## Methods

**Construction of mutants, protein expression, refolding, and purification.** Constructed plasmids of site-directed mutants of *P. patens* PsbS were purchased from BaseClear B.V.®. Single mutants were constructed in which E71 was replaced by Q (E71Q) or E176 was replaced by Q (E176Q) that are referred to as M1 and M2, respectively. A double mutant in which both E71 and E176 were mutated to Q (E71Q/E176Q) was constructed and is referred to as M1/M2. The plasmids were transformed, and the mutant proteins were overexpressed in *E. coli* and purified as has been described for WT PsbS in ref. [21]. Briefly, the mutant *P. patens* PsbS genes were inserted into a pExp5-vector containing an N-terminal His$_6$-tag using Gibson assembly technique and overexpression of the target proteins was carried out in *E. coli* strain BL21(DE3) pLysS. Cultures were harvested 12–14 h after induction with isopropyl-ß,D-thiogalactopyranoside (ITPG) at cell density of 0.3–0.4 and cell pellets were stored at −80 °C until further use. For the NMR experiments, $^{13}$C and $^{15}$N uniform labeling of WT PsbS and M1/M2 mutant was carried out by protein overexpression using standard minimal media containing $^{13}$C-glucose and $^{15}$N-ammonium chloride. For purification, cell pellets were washed in buffer containing Triton X100, yielding white precipitates containing unfolded PsbS as inclusion body pellets. The inclusion body pellets were dissolved and incubated in urea buffer (25 °C, 900 rpm shaking for 30 min) followed by centrifugation at 20,000 x $g$ for 10 min. These steps were repeated twice and followed by washing in urea buffer containing 0.05% lithium dodecyl sulfate (LDS) to separate PsbS as pellet from impurities that were contained in the soluble fraction. Finally, the PsbS pellet was dissolved by adding 0.5% LDS and buffer exchange was performed to remove the high concentration of urea and adjust the pH.

WT and mutant PsbS were refolded in n-Dodecylphosphocholine (FC-12) detergent buffers at pH 5.0 and pH 7.5 conditions, using 100 mM sodium acetate (pH 5.0) or sodium phosphate (pH 7.5) buffers. For the refolding step, unfolded protein (~1 mg/mL) was mixed with an equal volume of refolding buffer and heated to 100 °C for 1 min after which FC-12 was added to the mixture and 200 mM KCl was used to precipitate and remove the LDS[21]. For the FTIR and 2DIR experiments, the proteins were prepared in deuterated detergent buffer equilibrated at pD 7.5 or 5.0. The pD was set with a standard pH meter using the relation pD = pH + 0.4, with pH the measured value on the pH meter.

**Sodium dodecyl sulfate (SDS)-page gel electrophoresis.** SDS-page gel electrophoresis analysis (12.5% running gel, 4% stacking gel stained with Coomassie brilliant blue R-250 Bio-Rad) was carried out for checking the yield of PsbS at every step of overexpression, purification, and refolding. For staining, 2.5 µL of Precision Plus Protein™ Dual Color Standard from Sigma was used.

**NMR spectroscopy.** Solid-state NMR measurements were performed on a Bruker Avance I 750-MHz wide-bore solid-state NMR spectrometer with 17.6 Tesla magnetic field. In this field, $^{13}$C and $^{1}$H resonate at 188.66 and 750.23 MHz, respectively. Standard 4 mm triple resonance MAS probe was used. All the samples were packed in 4 mm zirconium rotors with top insert and were spun at the magic

angle (54.74⁰). The spinning frequency was set at 13 kHz. The temperature was set at 293 K. $^{13}$C spectra were obtained through direct polarization also referred to as 'hpdec'. 90⁰ or (π/2) carbon pulses of 6.2 µs and 3.1 µs proton pulses were applied. An acquisition time of 36.2 ms was used. For the experiments, 1024 scans were acquired with a constant recycle delay of 5 s. The presented spectra contain the sum of 1024 accumulated scans and a line broadening function of 50 Hz was applied for processing. All the $^{13}$C spectra were externally referenced to methyl signal of tetramethylsilane (TMS).

**FTIR spectroscopy.** Infrared spectra were recorded using an FTIR spectrometer (IFS 66 s Bruker) equipped with a nitrogen-cooled photovoltaic Mercury Cadmium Telluride (MCT) detector (20 MHz, KV100, Kolmar Technologies) described earlier[55–57]. The samples were contained between CaF$_2$ windows separated with a 20 µm Teflon spacer and the concentration was tuned for OD ~0.8 absorption at 1650 cm$^{-1}$, at a protein concentration of approximately 1 mM, with a volume of 100 µl. The measurements were carried out at room temperature and spectral resolution of the instrument was 3 cm$^{-1}$. The samples were solubilized in D$_2$O detergent buffer at either pD 5.0 or pD 7.5. Measurements with the pD 5.0 and 7.5 samples were collected consecutively and difference FTIR spectra were produced by scaling according to the integrated linear absorption of the amide I band (1610–1690 cm$^{-1}$).

**2DIR spectroscopy.** Two-dimensional infrared spectroscopy (2DIR) was carried out in the pump-probe geometry using an acoustic optical modulator-based pulse shaper[58,59], as an extension to an existing femtosecond mid-IR spectrometer[45,60,61]. A 1 kHz Ti:sapphire amplifier (Spitfire Ace, Spectra Physics) pumped an optical parametric amplifier (Topas, Light Conversion) producing signal and idler beams that underwent subsequent difference frequency generation in a AgGaS$_2$ crystal yielding femtosecond midIR pulses tunable from 1–10 µm center wavelength with ~100 fs pulse duration. A 5% fraction was split off for the probe beam with a wedged CaF$_2$ window. The pump beam was modulated with a germanium acousto-optic modulator (Quickshape, Phasetech, Madison WI). Data was collected in the time domain by oversampling along the free induction decay (τ) in the rotating frame with a two-pulse, four-frame phase-cycling scheme to isolate the χ$^{(3)}$ response and remove contributions from scattered pump light in situ. The pump and probe beams were focused on the sample with an off-axis parabolic mirror, overlapped in space, and set to a designated delay using a motorized linear delay stage. The transmitted probe beam was collimated and detected with a 64-element Mercury Cadmium Telluride (MCT) photodiode array (Infrared Associates). A parallel polarization condition <ZZZZ> was employed by tuning the polarization of the pump beam with a combination λ/2 MgF$_2$ waveplate (Karl Lambrecht) and wire-grid polarizer (Thorlabs). The data of Fig. 4 were taken at $T = 400$ fs. The PsbS samples were contained between two CaF$_2$ windows separated with a 20 µm Teflon spacer, with an OD of 0.8 at 1650 cm$^{-1}$, at a protein concentration of approximately 1 mM, with a volume of 12 µl. D$_2$O buffer solution was used to minimize the background absorption. Measurements with the pD 5.0 and 7.5 samples were collected consecutively and difference 2DIR spectra were produced by scaling according to the integrated linear absorption of the amide I band (1610–1690 cm$^{-1}$).

**Reporting summary.** Further information on research design is available in the Nature Research Reporting Summary linked to this article.

## Data availability

The main data supporting the findings of this study have been deposited in repository DataverseNL with identifier https://doi.org/10.34894/SVGHD4. The additional data are available from the corresponding authors on reasonable request.

## Code availability

NMR and FTIR data were taken with commercial Bruker software. 2DIR data were taken with commercial PhaseTech software. Custom LabView code were developed as part of this study for use in conjunction with code supplied by PhaseTech. Details of modifications are available from the corresponding authors upon reasonable request.

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

## Acknowledgements

Dr. Karthick Babu Sai Sankar Gupta is greatly acknowledged for assistance with the NMR spectroscopy. This work was supported by the Chemical Sciences Council of the Netherlands Organization for Scientific Research (NWO) through a NWO-CW VICI grant to J.T.M.K (nr. 724.011.004). and an NWO-CW VIDI grant to A.P. (nr. 723.012.103).

## Author contributions

P.K, J.T.M.K., and A.P. designed research, P.K and M.K. performed research, M.K. and A.P. contributed new reagents or analytic tools, P.K., M.K., J.T.M.K., and A.P. analyzed data, P.K., J.T.M.K., and A.P. wrote the paper.

## Competing interests

The authors declare no competing interests.
