## [Peer Review File · Nature Communications]

Reviewers' comments:

Reviewer #1 (Remarks to the Author):

This manuscript describes the pH-dependent response mechanism of an thylakoid-intrinsic protein of photosystem II, with some parts of it facing the lumen and some the stroma. It is involved in the non-photochemical quenching and creation of LHCII and PSII supercomplexes. The main methodologies used is solid state NMR, which is appropriate for a membrane protein, and 2DIR. The results reported here have apparently been acquired in the framework of the PhD thesis "A study on PsbS and its role as a pH sensor" by M. Krishnan, who is first author of the submitted work. Other results from this thesis, which is accessible online (<https://openaccess.leidenuniv.nl/handle/1887/76580>) have been reported in a previous publication (Scientific Reports 7: 15200, DOI:10.1038/s41598-017-15068-3).

The experimental work appears to be done with state of the art methodology and the paper is scientifically sound.

The conclusions based on the experimental results propose ne mechanisms for the action of PsbS in vivo. However, the interpretation still seems to be speculative in parts and more structural proof - in particular concerning the roles and locations of helices H2 and H3 with respect to the membrane would be desirable. (see below)

There are a few question and suggestions, which I would suggest to be addressed by the authors before publication:

(1) Did the authors consider adding an additional pH step between the 7.5 and 5.0 which could provide a cue to the transitional state between the two conformations? This could actually disambiguate the interpretation of Fig. 2, and one could probably reveal in which direction the two peaks are shifting, instead of assuming that they are buried beneath the other signals.

(2) The author compare spectra at two pH values, but they use two different buffer systems (acetate and phosphate) - did they consider possible influence of the buffer anions on the stability of the two conformations?

(3) Were any multidimensional NMR experiments performed? For instance, 2D DARR could reveal how many amino acids are affected by structural changes.

(4) In spite of the expected difficulties with resonance assignment of this protein, has at least a partial assignment been attempted?
A valuable support of the structural interpretation could be achieved by selective isotopic labelling (or unlabeled). This would allow one to really discriminate signals of the key amino acids.

(5)Have you considered that the switch mechanism proposed might be temperature dependent. Could this be checked by different methods, e.g. solution NMR, CD?

(6) This reviewer is not an expert in 2DIR, however, in particular in view of the fact that 2DIR is not a very common approach, the interpretations of the 2DIR spectra in the paper would need some further clarification.

(7) Minor point: Addition of a protein size "ladder" into the figures of SDS-PAGE gels would be beneficial.

In summary, this manuscript contains some valuable new ideas concerning the mechanism of PsbS

pH response, which are for sure of interest for the PSII/LHC research field.

Reviewer #2 (Remarks to the Author):

The study by Krishan, et al. seeks to elucidate the molecular mechanism of pH activation of the sensor protein PsbS. The mechanisms underlying the stress responses of photosynthesis are critical outstanding questions in the field. A better understanding would have substantial and broad impact. In principle, the study should be published in a high impact journal like Nature Communications. At this point, however, there appear to be issues with the 2D IR spectra on which the much of the conclusions and discussion are based. I cannot recommend publication in the current form.

Major concern: The 2D peaks do not always correspond to the 1D spectrum or the text of the manuscript (starting on pg 10). The 1D and 2D absorptions should appear at the same frequencies (or nearly the same, accounting for effects due to differing transition dipole dependence and overlapping bands). Most concerning, in Figure 4B, I see diagonal bands at ~ 1639 , 1645 , and 1665 cm^{-1} . There is no amplitude at 1630 cm^{-1} , which is discussed extensively in the text. In contrast, the 1D data in Figure 4B show a large bleach at 1622 cm^{-1} and smaller bleaches at ~ 1640 and 1685 cm^{-1} . (I have attached a marked up copy of Figure 4 to illustrate my concern.) Figure 4C shows a 2D band at 1630 cm^{-1} and a qualitatively distinct 2D spectrum than shown in Figure 4B. In contrast, the 1D difference spectra of the WT and M2 mutant qualitatively look very similar; the main differences appears to be the magnitude of the bleach at 1622 cm^{-1} . Figure 4A shows substantial signals off the diagonal at the excitation frequency of 1670 cm^{-1} that are not accounted for.

Moderate concerns/comments:

- Examples of all absolute 2D IR spectra should be provided at least in SI.
- Determination of inhomogeneous broadening in the presence of many overlapping bands by simple inspection of the nodal slope is not likely to give accurate results. There are a couple papers in the literature (see M. Fayer group publications) to guide analysis when multiple overlapping bands are present.
- The manuscript states in several places that the pH dependent conformational changes are suppressed in the M2 mutant. The difference 2D spectra in Fig. 4C do show a loss of species with signals at 1630 and 1647 cm^{-1} . Technically the data do suggest a pH induced conformational change for M2, just a different one than for WT. The term "locked" for M2 does not seem appropriate.
- On page 13, the manuscript states, "This assignment is consistent with the FTIR amplitude of the difference signal in WT PsbS, corresponding to at least 8 amino acids..." The basis for this quantitative comparison should be explicitly laid out.
- The language used to describe the 2D IR spectra on pg. 9 is not exactly correct. The negative bands on the diagonal result from not just ground state bleaching but also excited state stimulated emission. Calling the horizontal axis ω_{emis} seems misleading as it reports both absorptive and emissive features. Referring to the excited state absorption signal which is shifted from the diagonal by the anharmonicity as a "satellite peak" also seems odd.

Minor concerns/comments:

- The statement on pg. 3, "2D infrared (2DIR) spectroscopic methods were developed..." with ref. 28 is misleading. The citation for 2D IR spectroscopy applied to proteins is one published in the most high impact journal, which is presumably why it was chosen. However, the cited paper used specific isotopic labeling to probe specific residues, so my expectations were at odds with the content of the paper following.
- It should be make clear whether the difference in the pH/pD was taken into account when designing the experiments to compare equivalent conditions.
- Plotting the 1D and 2D spectra (Figs 3 and 4) in the same manner (low to high freq vs high to

low) would be preferable.

-I discern a sharp derivative shape in the difference spectra of Figure 3 around 1710 cm⁻¹, rather than an induced absorption as interpreted by the authors.

-Rather than renaming the mutated proteins as M1, M2, and M3, it seemingly would be clearer to refer to them as E71Q, E176Q, and E71Q/E176Q. M2 is the only one extensively discussed and the renaming only reduces the name by one letter. Similarly, Glu-1 and Glu-2 are referred to in the main text instead of E71 and E176. To me, just calling residues by the original names would be clearer for readers.

Reviewer #3 (Remarks to the Author):

This manuscript describes NMR and IR spectroscopy experiments to investigate conformational changes in the PsbS protein at low pH. How PsbS responds to low thylakoid lumen pH to induce non-photochemical quenching (NPQ) of the energy-dependent type (qE) is an important, unresolved question in chloroplast biology. The results show that there are indeed changes in the secondary structure of refolded and detergent-solubilized recombinant PsbS upon protonation of glutamate and aspartate residues in the protein, as expected. The difficulty is in assigning the spectroscopic changes to specific changes in secondary structure within the protein. Based on site-directed mutants affecting two key lumen-facing glutamate residues that are important for PsbS function *in vivo*, the authors assign spectroscopic changes to conformational changes in two short amphipathic helices, H2 and H3. These assignments are hypothetical but plausible, however the results are not consistent with previous site-directed mutagenesis *in vivo* or molecular dynamics simulations *in silico*.

1. As stated at the top of p. 6, the NMR results are inconsistent with the recent MD results of Liguori et al. (2019, JPCL). These discrepancies should be addressed and discussed.

2. At both pD 7.5 and 5.0, the M2 mutant of PsbS affecting Glu-2 (with the E176Q change) appears to have FTIR signals at 1625-1630 cm⁻¹ and 1660 cm⁻¹ that resemble those of the WT protein at pD 5.0, suggesting that the protein is already in a low pH conformation even at pH 7.5. If this interpretation is correct, then the M2 mutant should be in a constitutively active (low pH) state, but this is inconsistent with NPQ measurements on this mutant *in vivo* that showed no constitutive NPQ at neutral pH and a defect at low pH/high light (Li et al., 2004, JBC).

3. The loss of the Amide I response in M2 is interpreted to show that amphipathic helix H2 is "likely" the site that undergoes pH-dependent structural change. Glu-2 is in helix H2, so that seems plausible. However, the statement (middle of p. 13) that the amplitude of the difference signal in WT (corresponding to at least 8 amino acids) is consistent with the length of the H2 helix (9 amino acids) seems like weak supporting evidence for this assignment. Those 8 amino acids could be anywhere in the protein and might not even be contiguous. Furthermore, MD simulations (Liguori et al., 2019, JPCL) suggest that helix H2 does not exhibit any pH-dependent changes, which is inconsistent with this assignment.

4. The manuscript focuses on the M2 mutant and does not really consider the M1 and M3 mutants, because the authors argue that these mutants adopt a non-native fold (top of p. 9). However, the logic for this argument (in the Fig. S3 legend) does not make sense to me. The authors state that the charge distribution along the first luminal loop near Glu-1 (E71) in M1 and M3 will differ from the WT at both neutral and low pH, but the exact same argument could be made for the second luminal loop in the M2 mutant. Conversely, the protonated carboxyl signals at 1700-1750 cm⁻¹ are used to support unaltered protonation states in M2 (middle of p. 8), but the same signals are also present in M1 and M3. The M1 and M2 mutants were shown to have a similar NPQ phenotype *in vivo*, and the M3 mutant eliminated qE (Li et al., 2004, JBC). This again seems inconsistent with the results of this manuscript.

5. A possible explanation for the discrepancies between the results in this manuscript and prior results is that the spectroscopic experiments described here were done on PsbS in detergent micelles. In previous experiments *in vivo*, PsbS was in the native thylakoid membrane of chloroplasts in intact leaves, and the experiments *in silico* were done on PsbS in a native-mimicking membrane. This important difference raises a question about the relevance of the results with detergent-solubilized PsbS protein to the situation in the thylakoid membrane *in vivo*, and this question should be acknowledged and discussed in this manuscript.

6. An FTIR band at 1550 cm^{-1} is described but not shown in Fig. 3. The x-axis range should be extended to lower wavenumbers to make this band evident.

7. Why was pH 5.0 selected as the low pH condition? pH 5.0 is below the physiologically relevant pH range within which PsbS seems to work *in vivo*, according to experiments that estimated the pK_a of PsbS *in vivo* (Takizawa et al., 2007 BBA).

8. The introductory text could be edited to make it more concise and precise. Specifically:

- p. 2, last sentence of first paragraph needs references.
- p. 2, second sentence of second paragraph should cite original paper rather than a review.
- p. 2, fourth sentence of second paragraph cites a long list of papers, many of which do not seem directly relevant to the statement in this sentence. Papers showing the pH sensing function of PsbS, interactions with antenna proteins, and reorganization of PSII should be cited here. This sentence is largely repeated at start of third paragraph.
- p. 2, sixth sentence of second paragraph omits the fact that VDE and ZEP were overexpressed in addition to PsbS in the cited paper (Ref. 20).

9. The Methods are lacking essential details that are necessary to allow the results to be replicated. For example, what concentration of detergent was used for refolding? What amount and concentration of protein sample was used for NMR and 2DIR spectroscopy?

Reviewer #4 (Remarks to the Author):

The present manuscript reports the infrared analysis of PsbS, a key protein in the photoprotection process of plant called non-photochemical quenching (NPQ). To clarify the molecular mechanism of the conformational pH switch of PsbS, the authors performed FTIR/2DIR measurements of the single and double mutants of two Glu residues (E71Q (M1), E176Q (M2), and E71Q/E176Q (M3)), which are the candidates of residues responsible for pH dependent conformational changes. NMR was also used but only to confirm pH-induced deprotonation/protonation of nonspecific carboxylate groups in the protein. They measured FTIR spectra of all the three mutants as well as the WT protein, but they focused on the analysis of amide I bands of WT and M2 to make a model of pH-dependent conformational changes of PsbS. 2DIR data were also obtained for WT and M2 to increase the resolution and provide the information of heterogeneity of bands. Although the proposed model is reasonable, assignments of the amide I peaks to the helical components (H2 and H3) and their conformational changes are rather speculative. In particular, the peak frequency of 1625-1630 cm^{-1} assigned to the H2 helical component is significantly low and usually in the β -sheet region, but could be possible in the case of some specific interaction of water as the authors suggested. However, without any conclusive evidence, the impression remains that the assignment of the IR bands are rather tentative. The 1625 cm^{-1} peak at pH 7.5 also diminished by M1 mutation near H3, which is at odds with the assignment to H2. This could be interpreted in the context of the interaction between H2 and H3 in a dimer or the reduced dimer population, but there is no explanation in the text. IR measurements using mutants of other carboxylate residues such as E78 and E162 (*P. patens*) will be useful as control experiments to remove the possibility

that the authors actually detected unspecific effects of mutations of carboxylate groups on the luminal side.

Other points:

1. The 2DIR map in Fig. 4A and B may be shifted to higher frequencies. The map in Fig. S5 is correct. All the 2DIR data of WT at pH 5 and the M2 mutant at pH 5 and 7.5 before taking subtraction should be presented at least in Supporting Information.
2. Fig. 6 is not based on the authors' data. It may be involved in the Fig. 1 as an introduction. The detail of the hydrogen bond interaction is not very clear without atomic color in the interacting residues shown in green and red.
3. E176(173) functions as a hydrogen bond acceptor in the dimer interaction. This interaction may be retained even in Q176 in M2. What is the reason for the decreased dimer population in this mutant? Also, mutation of E71(69) to Q may not interfere with the H2-H3 interaction at low pH, because it is not charged. Please explain the reason for the effect of these mutations on dimerization.
4. p. 8, last paragraph: No NMR data of M3 are shown in Fig. 2.
5. p. 20, line 10: Fig. 6 may be Fig. 4.

Reviewer #1 (Remarks to the Author):

This manuscript describes the pH-dependent response mechanism of an thylakoid-intrinsic protein of photosystem II, with some parts of it facing the lumen and some the stroma. It is involved in the non-photochemical quenching and creation of LHCII and PSII supercomplexes. The main methodologies used is solid state NMR, which is appropriate for a membrane protein, and 2DIR.

The results reported here have apparently been acquired in the framework of the PhD thesis “A study on PsbS and its role as a pH sensor” by M. Krishnan, who is first author of the submitted work. Other results from this thesis, which is accessible online (<https://openaccess.leidenuniv.nl/handle/1887/76580>) have been reported in a previous publication (Scientific Reports 7: 15200, DOI:10.1038/s41598-017-15068-3).

The experimental work appears to be done with state of the art methodology and the paper is scientifically sound.

The conclusions based on the experimental results propose ne mechanisms for the action of PsbS in vivo. However, the interpretation still seems to be speculative in parts and more structural proof - in particular concerning the roles and locations of helices H2 and H3 with respect to the membrane would be desirable. (see below)

There are a few question and suggestions, which I would suggest to be addressed by the authors before publication:

(1) Did the authors consider adding an additional pH step between the 7.5 and 5.0 which could provide a cue to the transitional state between the two conformations? This could actually disambiguate the interpretation of Fig. 2, and one could probably reveal in which direction the two peaks are shifting, instead of assuming that they are buried beneath the other signals.

PsbS switches between two distinct states, so we do not expect any meaningful new information from choosing a pH value in between. The FTIR data fully agrees with the NMR data in showing protonation of all titratable residues at pH 5.0. Because the experimental steps for each sample type are very challenging and time consuming, i.e. production of mutants, refolding of the protein in detergent buffer, biochemical characterization, NMR spectroscopy, optimizing settings for FTIR, performance and analysis of the 2DIR experiments, we limited this study to two pH conditions.

(2) The author compare spectra at two pH values, but they use two different buffer systems (acetate and phosphate) - did they consider possible influence of the buffer anions on the stability of the two conformations?

We have to change the buffer system to stabilize the protein at pH 5.0 conditions. We do not expect large effect of the buffer anions on the conformation, considering that the proteins are

surrounded by detergent micelles. In fact, varying the type of detergents does have some effect on the spectral shape of the CD spectrum of PsbS¹ while the helical content remains the same and the effect is similar at neutral and low pH conditions. The spectral variation in the CD spectra for different detergent buffers was ascribed to changes in the coiled stromal loops of PsbS.

(3) Were any multidimensional NMR experiments performed? For instance, 2D DARR could reveal how many amino acids are affected by structural changes.

Unfortunately, samples of PsbS in detergent micelles are not suitable systems for multidimensional solid-state NMR; samples at room temperature are liquid-like while the NMR resolution of frozen detergent solutions is poor. For 2D DARR experiments, we would have to prepare PsbS proteoliposomes. We started pilot NMR experiments on those, which need to be refined because of the low stability of proteoliposomes when they are loaded with high amounts of PsbS. Another direction to go for would be solution NMR, performing TROSY experiments on deuterated samples of PsbS in detergent micelles. We analyzed HSQC spectra of PsbS in different types of detergents to find the most suitable detergent conditions for solution NMR experiments¹ and may pursue in this direction.

(4) In spite of the expected difficulties with resonance assignment of this protein, has at least a partial assignment been attempted?

A valuable support of the structural interpretation could be achieved by selective isotopic labelling (or unlabeled). This would allow one to really discriminate signals of the key amino acids.

A partial NMR assignment has not been performed yet for reasons mentioned above. Indeed, selective isotope labeling or un-labeling will be a valuable support here. However, selective Glu labeling using host organisms is notoriously difficult due to label scrambling via metabolic pathways. To circumvent this problem, we already set up a cell-free expression system for production of PsbS.² However, because of the technical challenges, selective isotope labeling and NMR assignments are considered outside the scope of this study.

(5) Have you considered that the switch mechanism proposed might be temperature dependent. Could this be checked by different methods, e.g. solution NMR, CD?

Below we include a set of CD experiments recorded between 5 and 45 degrees Celcius collected of wildtype PsbS in FC-12 detergent micelles at pH 7.5 condition (Fig.1) and NMR HSQC spectra of PsbS at pH 5.0 and at pH 7.5, recorded at two temperatures (20 °C and 40 °C, Fig.2 and 3). For Fig. 1, we were not able to gather the original data from the CD setup to make a suitable figure due to Covid-19 lab access restrictions at this moment. We apologize therefore to show a photograph of a lab-book picture instead.

The CD and NMR spectra do not show any temperature-dependent conformational changes.

The HSQC spectra at pH 5.0 have more resolved signals and a differential pattern compared to the spectrum at pH 7.5 at both temperatures. Several additional signals are resolved at the higher temperature, owing to increased protein dynamics at high T.

Figure 1 CD spectra collected of wildtype PsbS in FC-12 detergent micelles at pH 7.5 conditions

Figure 2 HSQC spectra of wildtype PsbS in FC-12 detergent micelles at pH 7.5 (red) and pH 5.0 (black), collected at 20 °C.

Figure 3 HSQC spectra of wildtype PsbS in FC-12 detergent micelles at pH 7.5 (red) and pH 5.0 (black), collected at 40 °C.

(6) This reviewer is not an expert in 2DIR, however, in particular in view of the fact that 2DIR is not a very common approach, the interpretations of the 2DIR spectra in the paper would need some further clarification.

To address the reviewer's concern, we clarified some of the basic features of 2DIR spectroscopy in the revised text, on p. 10, middle, p. 11, bottom, p. 12, top, and added a citation to a comprehensive textbook by Hamm and Zanni.³ We emphasize that 2DIR spectroscopy has been applied to polypeptides and (small) proteins for two decades now, and it may be regarded as an established technique.

(7) Minor point: Addition of a protein size "ladder" into the figures of SDS-PAGE gels would be beneficial.

We showed a protein-size ladder, but did not add the protein sizes to the ladder. We apologize for this and added those to Figure S1.

In summary, this manuscript contains some valuable new ideas concerning the mechanism of PsbS pH response, which are for sure of interest for the PSII/LHC research field.

We highly appreciate the reviewer's positive evaluation our work.

Reviewer #2 (Remarks to the Author):

The study by Krishan, et al. seeks to elucidate the molecular mechanism of pH activation of the sensor protein PsbS. The mechanisms underlying the stress responses of photosynthesis are critical outstanding questions in the field. A better understanding would have substantial and broad impact. In principle, the study should be published in a high impact journal like Nature Communications. At this point, however, there appear to be issues with the 2D IR spectra on which the much of the conclusions and discussion are based. I cannot recommend publication in the current form.

Major concern: The 2D peaks do not always correspond to the 1D spectrum or the text of the manuscript (starting on pg 10). The 1D and 2D absorptions should appear at the same frequencies (or nearly the same, accounting for effects due to differing transition dipole dependence and overlapping bands). Most concerning, in Figure 4B, I see diagonal bands at ~1639, 1645, and 1665 cm^{-1} . There is no amplitude at 1630 cm^{-1} , which is discussed extensively in the text. In contrast, the 1D data in Figure 4B show a large bleach at 1622 cm^{-1} and smaller bleaches at ~1640 and 1685 cm^{-1} . (I have attached a marked up copy of Figure 4 to illustrate my concern.)

We thank the reviewer for pointing out this frequency mismatch. By accident, a plot was made for Fig. 4 for the submitted manuscript with the wrong frequency calibration. We apologize for this oversight and applied the correct calibration in the revised manuscript. So, this frequency mismatch has been fixed in the updated plot and should resolve the reviewer's concern.

Figure 4C shows a 2D band at 1630 cm^{-1} and a qualitatively distinct 2D spectrum than shown in Figure 4B. In contrast, the 1D difference spectra of the WT and M2 mutant qualitatively look very similar; the main differences appears to be the magnitude of the bleach at 1622 cm^{-1} .

We respectfully disagree with the reviewer that the response in the M2 mutant is similar to that of WT. The FTIR difference spectra indicate that the M2 mutant has a significantly diminished amplitude of the conformational changes as compared to WT. The amplitude is a measure for the *absolute extent* of structural changes and for that reason, we proposed that the conformational changes that occur in WT are *largely* suppressed in the M2 mutant.

There is a residual low-amplitude response in the M2 mutant that we do not interpret in detail: the most significant feature in the M2 FTIR difference spectrum (Fig. 3B) is a bleach at 1625 cm^{-1} , (much smaller than in WT) accompanied by a very small negative/positive feature at 1643/1655 cm^{-1} . We believe the bleach at 1625 cm^{-1} is genuine and it likely represents a small, uncharacterized change in a solvent-exposed amphipathic helix, i.e., of H1 or H2. The very small negative/positive feature at 1643/1655 cm^{-1} is so small that it at most represents a change of a single amino acid. It has in fact a different spectroscopic signature than in wild type: in the latter, no negative signal and only a positive band around 1660 cm^{-1} is observed.

In the 2DIR difference spectrum of M2 (Fig. 4C), the signal amplitude is also diminished with respect to WT (See fig. S4 where diagonal slices are shown for WT and M2). Here, the difference signal is dominated by the positive band at 1625 cm^{-1} with accompanying anharmonic negative band, which coincides with the 1625 cm^{-1} bleach observed in FTIR. The remainder of the 2DIR difference map shows a very small positive diagonal band around 1645 cm^{-1} , which in combination with the FTIR results, is assigned to a minor unfolding or destabilization of a transmembrane helix by one amino acid, as judged from its amplitude with respect to the total Amide I absorption. We emphasize that in 2DIR, the distinct negative 1638 cm^{-1} diagonal band that was observed in WT and assigned to refolding of H3 is entirely missing in the M2 mutant. **We added a paragraph on p. 17 to discuss this issue.**

Figure 4A shows substantial signals off the diagonal at the excitation frequency of 1670 cm^{-1} that are not accounted for.

The reviewer raises a valid point here. **We added our interpretation of this off-diagonal band on p. 14, top, of the revised manuscript**

Moderate concerns/comments:

-Examples of all absolute 2D IR spectra should be provided at least in SI.

Is the reviewer referring to absolute value or (absorptive) equilibrium spectra? Regarding the former, there are no phase corrections required in our 2DIR experimental geometry. As such, we are guaranteed to retrieve the correct absorptive lineshape for each of the spectra reported.

For the latter, we have presented an example of the steady-state 2DIR spectrum in Figure 4A. Because the small magnitude of the pH-dependent signal changes, there is little value including equilibrium spectra for both states in the main text. **However, we have now provided these in the Supporting Information for completeness (Fig. S4).**

-Determination of inhomogeneous broadening in the presence of many overlapping bands by simple inspection of the nodal slope is not likely to give accurate results. There are a couple papers in the literature (see M. Fayer group publications) to guide analysis when multiple overlapping bands are present.

We understand the reviewer's concern: a quantitative scheme for calculating the nodal line slope (NLS) of a two-component system is given in a paper by Fayer et al,⁴ the main conclusion being the NLS is a weighted average of constituent bands. However, this formula is not valid in our case given there are more than 2 overlapping bands among the three substates in question. We emphasize that our conclusions do not rest on a quantitative determination of the NLS, but rather the qualitative relative difference in inhomogeneous broadening. Since the extent of overlap is roughly equivalent in all cases and the change in NLS is easily discernable, we feel this qualitative comparison is valid.

-The manuscript states in several places that the pH dependent conformational changes are

suppressed in the M2 mutant. The difference 2D spectra in Fig. 4C do show a loss of species with signals at 1630 and 1647 cm^{-1} . Technically the data do suggest a pH induced conformational change for M2, just a different one than for WT. The term “locked” for M2 does not seem appropriate.

We agree with the reviewer that the changes around 1625 cm^{-1} in the M2 mutant do not entirely disappear in FTIR and 2DIR, but they are of a significantly smaller amplitude than in wild type, as we stated in response to a comment above. That’s why we state that these changes are ‘largely suppressed’ in M2 (p. 12, bottom) and we call it a ‘pseudo wild type low pH conformational state’ (p. 21, middle). The distinct conformational changes in WT that involve H2 and H3 do not occur in the M2 mutant, as the pronounced negative diagonal signals at 1638 and 1666 cm^{-1} in WT are clearly absent in the M2 mutant. As we explained in one of the points above, we assign the residual signals in the M2 mutant to a small, further uncharacterized motion of one of the amphiphatic helices, and the signals around 1650 cm^{-1} as a minor unfolding/destabilization of a transmembrane helix. **As for one of the points above, the added text on p. 17 is relevant to this comment. To further satisfy the reviewer, we replaced the term ‘locked’ with ‘resides in’ on p. 8 and 21, so as make the formulation more accurate.**

-On page 13, the manuscript states, “This assignment is consistent with the FTIR amplitude of the difference signal in WT PsbS, corresponding to at least 8 amino acids...” The basis for this quantitative comparison should be explicitly laid out.

The reviewer may have missed this, but we already explicitly explained this on p. 8, middle : ‘The integrated bleach amplitude of the 1625 cm^{-1} band corresponds to 3.6% of the integrated Amide I absorption, and with a total of 221 amino acids in *P. patens* PsbS, hence involves at least 8 amino acid backbone oscillators, indicating that a conformational change occurs involving at least 8 amino acid residues’.

-The language used to describe the 2D IR spectra on pg. 9 is not exactly correct. The negative bands on the diagonal result from not just ground state bleaching but also excited state stimulated emission.

Indeed, the nonlinear response contains contributions from ground state bleach, stimulated emission, and excited state absorption. **We have amended the text to indicate this on p. 10, middle.**

Calling the horizontal axis ω_{emis} seems misleading as it reports both absorptive and emissive features.

We appreciate the reviewer’s concern in the light of the previous comment, but the notation of ω_{exc} and ω_{emis} is standard convention among the 2D spectroscopy community, which reflect the excitation/emission frequency of a given oscillator with respect to a fixed waiting time (T). This language is distinct from the ‘absorptive’ component of the nonlinear response which reflects loss or gain of excited state population.

Referring to the excited state absorption signal which is shifted from the diagonal by the anharmonicity as a “satellite peak” also seems odd.

We deleted the term ‘satellite peak’ and now speak of ‘corresponding anharmonic peak’.

Minor concerns/comments:

-The statement on pg. 3, “2D infrared (2DIR) spectroscopic methods were developed...” with ref. 28 is misleading. The citation for 2D IR spectroscopy applied to proteins is one published in the most high impact journal, which is presumably why it was chosen. However, the cited paper used specific isotopic labeling to probe specific residues, so my expectations were at odds with the content of the paper following.

We chose ref. 28 because that work is one of the very few papers where 2DIR spectroscopy was applied to a membrane protein (the well-known ion channel KcsA): the vast majority of 2DIR work has been done on small polypeptides and small water-soluble proteins. Since our work is rather unique as it concerns a 2DIR study on a membrane protein (and a very difficult one at that!), we felt it was appropriate to cite ref. 28 even if we did not apply the extensive isotope labeling in that study. **Nevertheless, prompted by the reviewer’s comment, we reformulated this sentence on p. 3, bottom to more accurately represent the approach that we took.**

-It should be make clear whether the difference in the pH/pD was taken into account when designing the experiments to compare equivalent conditions.

We conducted the discussion in terms of pD, which implicitly means that the difference between pH and pD has been taken into account. **We now explicitly mention this in the Methods section on p. 23, middle.**

-Plotting the 1D and 2D spectra (Figs 3 and 4) in the same manner (low to high freq vs high to low) would be preferable.

We understand the reviewer’s viewpoint, but we prefer to maintain the conventions used in the FTIR and 2DIR communities, which are opposite, unfortunately. In Fig. 4B,C the FTIR and 2DIR spectra are plotted together, so there they can be directly compared.

-I discern a sharp derivative shape in the difference spectra of Figure 3 around 1710 cm^{-1} , rather than an induced absorption as interpreted by the authors.

We presume that the reviewer refers to the slight derivative-like signal around 1690 cm^{-1} in WT. This represents a rather low-amplitude modulation of the larger positive induced absorption signal caused by carbonylic acid protonation signal at low pD. The signal remains all-positive in that region. We currently do not know the origin of the modulation. **We added a remark to the revised manuscript to indicate this on p. 9, top.**

-Rather than renaming the mutated proteins as M1, M2, and M3, it seemingly would be clearer to refer to them as E71Q, E176Q, and E71Q/E176Q. M2 is the only one extensively discussed and the renaming only reduces the name by one letter. Similarly, Glu-1 and Glu-2 are referred to in the main text instead of E71 and E176. To me, just calling residues by the original names would be clearer for readers.

We understand the reviewer's point, but we have adopted this nomenclature so as to avoid confusion between the different PsbS amino acid residue numberings in spinach (of which the X-ray structure has been determined), *Physcomitrella patens* (this work) and *Arabidopsis thaliana* (on which many of the *in vivo* studies have been performed).

Reviewer #3 (Remarks to the Author):

This manuscript describes NMR and IR spectroscopy experiments to investigate conformational changes in the PsbS protein at low pH. How PsbS responds to low thylakoid lumen pH to induce non-photochemical quenching (NPQ) of the energy-dependent type(qE) is an important, unresolved question in chloroplast biology. The results show that there are indeed changes in the secondary structure of refolded and detergent-solubilized recombinant PsbS upon protonation of glutamate and aspartate residues in the protein, as expected. The difficulty is in assigning the spectroscopic changes to specific changes in secondary structure within the protein. Based on site-directed mutants affecting two key lumen-facing glutamate residues that are important for PsbS function *in vivo*, the authors assign spectroscopic changes to conformational changes in two short amphipathic helices, H2 and H3. These assignments are hypothetical but plausible, however the results are not consistent with previous site-directed mutagenesis *in vivo* or molecular dynamics simulations *in silico*.

We respectfully disagree with the remark of the reviewer our assignments are hypothetical, as we will explain below. Also, we disagree with the reviewer's remark that our results are not consistent with previous studies. We present experimental results on isolated protein that are *complementary* to *in-vivo* studies and MD simulations. Molecular models for activation of PsbS have been proposed on basis of *in-vivo* studies, before detailed molecular knowledge of the system was available. We discuss the relationship between current results, *in vivo* mutagenesis and MD simulations in our answers below.

1. As stated at the top of p. 6, the NMR results are inconsistent with the recent MD results of Liguori et al. (2019, JPCL). These discrepancies should be addressed and discussed.

We have done so, as outlined below in more detail.

2. At both pD 7.5 and 5.0, the M2 mutant of PsbS affecting Glu-2 (with the E176Q change) appears to have FTIR signals at 1625-1630 cm⁻¹ and 1660 cm⁻¹ that resemble those of the WT protein at pD 5.0, suggesting that the protein is already in a low pH conformation even at pH 7.5. If this interpretation is correct, then the M2 mutant should be in a constitutively active (low pH) state, but this is inconsistent with NPQ measurements on this mutant *in vivo* that showed no constitutive NPQ at neutral pH and a defect at low pH/high light (Li et al., 2004, JBC).

The reviewer raises an important point here. We believe that it is very hard to establish a direct one-to-one relation between a singular molecular membrane constituent and a specific *in vivo*

response, especially in the case of PsbS and the thylakoid membrane. Our results suggest that there are multiple layers of interaction between molecular switching of PsbS and *in vivo* response. Thus far, most of our understanding of PsbS action has been top-down, on the thylakoid membrane level. There is a long way to go before we understand the precise mechanism by which PsbS triggers NPQ from the bottom up, in a systems level response. Based on the PsbS X-ray structure, this work constitutes a first experimental characterization of the molecular pH response of PsbS, a starting point from which we aspire to further build a true molecular mechanistic model of PsbS activation. We hope that the reviewer understands that this will be a multi-step process that will take many years and will involve many research groups, and that this work will not provide the final word on this important topic.

We have added a paragraph on p. 21 to discuss the relation between our observations and the known *in vivo* responses.

3. The loss of the Amide I response in M2 is interpreted to show that amphipathic helix H2 is “likely” the site that undergoes pH-dependent structural change. Glu-2 is in helix H2, so that seems plausible. However, the statement (middle of p. 13) that the amplitude of the difference signal in WT (corresponding to at least 8 amino acids) is consistent with the length of the H2 helix (9 amino acids) seems like weak supporting evidence for this assignment. Those 8 amino acids could be anywhere in the protein and might not even be contiguous.

We respectfully disagree with the reviewer on this point. We note that these 8 amino acids absorb at the same specific vibrational frequency at $1625 - 1630 \text{ cm}^{-1}$, which is unusually low for a helical element. It demonstrates that this signal represents a helical element that is solvent-exposed, so the reviewer’s statement that they could be anywhere in the protein is not correct. Thus, 2D-IR indicates that this signal originates from a helical fragment, which excludes the possibility of a β -sheet or loop. The frequency is much too low to be assigned to a transmembrane helix and is only consistent with a water-exposed amphipathic helix fragment. Taking the known structure of PsbS (and in fact of all LHC-type protein complexes), the only amphipathic helix fragments are those at the lumen, i.e. H1 and H2. Moreover, we show that mutation of Glu-2, which is located in H2, abolishes the signal. The amplitude matches with a stretch of 8-9 amino acids, which closely agrees with H2 but not with H1. Hence, we confidently assign this signal to H2. **We modified the text to emphasize this on p. 14, bottom and p. 15, top.**

Furthermore, MD simulations (Liguori et al., 2019, JCPL) suggest that helix H2 does not exhibit any pH-dependent changes, which is inconsistent with this assignment.

We respectfully disagree with the statement that our results are inconsistent with those of Liguori *et al*, and believe that this issue should be viewed in a broader context. Our results show that pH-dependent motion of the H2 helix is a solidly assigned, *intrinsic* property of PsbS, and the real question is under what circumstances this motion occurs and what its functional significance is. The H2 motion was revealed by us in PsbS in detergent micelles; the MD simulation was conducted on PsbS embedded in POPC lipid bilayer. It is important to note that such lipids make the system very flat and planar. The reviewer appears to suggest that the MD simulations mimic the *in vivo* situation, but this is hardly the case. Both model systems are to a certain extent different from PsbS in the thylakoid membrane, which consists of MGDG, DGDG, SQDG and

PG lipids that induce more membrane pressure and curvature (the most abundant thylakoid lipid, MGDG, is a non-bilayer lipid) and where the presence of LHC complexes causes a crowded environment with many protein-protein contacts. Water access to the luminal side may be somewhat better mimicked in the lipid bilayer model, but protein-protein interactions are present in our system as part of the proteins are dimerized (the MD simulations were conducted on a monomer). In that sense, both conditions are relevant, they are complementary and both offer valuable insights and should not be contrasted against one another. **We added a section to the manuscript on p. 19/20 to discuss this issue.**

Having said that, another limitation in the MD simulation that may have prevented it from reproducing the H2 repositioning at lower pH lies in the limited time window after pH change in the simulation, which was 4.7 μ s. It is well known that stimulus-induced conformational changes in proteins may take place on much longer timescales than that, i.e. from sub-millisecond to milliseconds. This is for instance the case for Photoactive Yellow Protein, where it was shown with NMR⁵ and IR spectroscopy⁶ that an N-terminal cap detaches and unfolds upon photoactivation, and that this process takes place on a millisecond timescale.⁷ Specific methods in MD such as transition path sampling and replica exchange are required to predict such slow dynamics.⁸ A similar case is presented by the LOV2 domain photoreceptor, where it was shown with NMR and IR spectroscopy that a C-terminal helix detaches and unfolds upon photoactivation,⁹⁻¹¹ and that this process takes place on a sub-ms to ms timescale.^{12,13} With MD simulations, only the very initial stages of C-terminal helix detachment and unfolding were resolved.^{14,15} Thus, if a MD simulation is not run over a sufficiently long time window, such conformational changes may be missed. **We made a remark on p. 19 to consider this issue.**

4. The manuscript focuses on the M2 mutant and does not really consider the M1 and M3 mutants, because the authors argue that these mutants adopt a non-native fold (top of p. 9). However, the logic for this argument (in the Fig. S3 legend) does not make sense to me. The authors state that the charge distribution along the first luminal loop near Glu-1 (E71) in M1 and M3 will differ from the WT at both neutral and low pH, but the exact same argument could be made for the second luminal loop in the M2 mutant.

We emphasize here that there is only a pseudo-C2 symmetry in PsbS, and that significant differences exist between the two halves of the protein that bind Glu-1 and Glu-2, respectively, especially at the luminal side. This fact needs to be appreciated for a proper understanding of our results.

The fold of the M2 mutant is close to that of WT at low pH as judged from the IR spectra. Instead, the IR spectrum of M1 and M3 does not resemble that of the wild type at either pH. The notion that M1 and M3 adopt a non-native fold is a conclusion derived from these experimental results and is not a theoretical argument. We provide a rationale for our results, by suggesting that the non-native fold could be due to a non-native charge distribution. At neutral pH, the charge E71 is eliminated by the mutation, but other protonatable residues in this loop (D68, E78 and E80 for *P. patens*) will be negatively charged, which gives a non-natural charge distribution along the loop as a whole (-4 in wild type, -3 in the M1 and M3 mutants).

Hence, contrary to the reviewer's statement, it is not true that the same argument holds for M2: at the Glu-2 site, there are no additional protonatable residues in the loop/H2 and elimination of the Glu-2 charge gives an overall charge of this loop that resembles the charge distribution of the wild type at low pH. As a consequence, the neutral pH and low pH structures may be similar, which we observe experimentally.

We add to this that mutations at the Glu-1 site are more likely to produce a non-natural fold compared to mutation at the Glu-2 site because of the differences in secondary structure between the two loops. According to the X-ray structure, the lumen-exposed loop containing Glu-1 is only locally structured, containing a short helix fragment H1 and a short 3_{10} helix fragment that carries Glu-1. 3_{10} helices are known to be intrinsically unstable, which is also demonstrated in the MD study of Liguori *et al.* where this fragment unfolds at high pH. A point mutation at Glu-1 therefore could easily induce an un-natural structural change. In contrast, the loop containing Glu-2 forms a well-defined structural element (i.e. an amphipatic helix of 9 amino acids) that will have a more stable fold and should be less affected by a point mutation.

We added an extensive section on p. 17-19 to clarify this issue.

Conversely, the protonated carboxyl signals at 1700-1750 cm^{-1} are used to support unaltered protonation states in M2 (middle of p. 8), but the same signals are also present in M1 and M3.

We think there is a misunderstanding here. Protonation of multiple carboxylates is seen as increase of signal in the region 1700-1750 cm^{-1} and decrease of signal in the region around 1570 cm^{-1} (deprotonated carboxylates). It should be noted that PsbS contains 19 titratable glutamates/aspartates and from our data, we conclude that they all protonate at pH 5.0. Hence, the NMR and FTIR signals will report on all of them, and deleting one or two glutamates will not affect these signals very much. **We should have noted that more clearly in the manuscript, and added sentences on p. 8, top and p. 9, middle to clarify this.**

The M1 and M2 mutants were shown to have a similar NPQ phenotype *in vivo*, and the M3 mutant eliminated qE (Li et al., 2004, JBC). This again seems inconsistent with the results of this manuscript.

Because the M1 and M3 mutant have a non-native fold, we decided to not further interpret the results of those mutants in detail as for any interpretation the link with the native protein would be speculative, as we describe on p. 19, middle of the revised manuscript.

The point raised by the reviewer may also be regarded from a reverse perspective: our results hint at the possibility that the impaired activity of the M1 and M3 mutants *in vivo* might be related to similar non-native folds of the PsbS mutant proteins in the thylakoid membrane rather than an effect of specific charge elimination of Glu-1.

The observation that the M1 and M2 mutants have a similar NPQ phenotype *in vivo* does by no means imply that their responses are similar at the molecular level. Such an idea arose at a time when PsbS was considered a C2 symmetric molecule and then seemed fully logical; we now know, with the advent of the X-ray structure, that only a pseudosymmetry applies and that significant differences exist between the local structures around Glu-1 and Glu-2. Hence, it is not

expected that the M1 and M2 mutants behave in a similar way on the molecular level. Yet, their disparate molecular responses may well result in a similar NPQ phenotype *in vivo*.

In our work, we identify specific responses for each of the active Glutamates Glu-1 and Glu-2. We observe pH-dependent changes in the position of H2 upon protonation of Glu-2 that could potentially change inter-protein hydrogen bond networks in a native membrane. Our work is complementary to the MD approach of Liguori *et al.*, who observed pH-dependent conformational changes due to protonation of Glu-1 that also potentially could affect inter-protein hydrogen bond networks (H3 refolding, of which we observe a spectroscopic signature in wild type PsbS). So, taking our results and that of Liguori together, both Glu sites have shown to undergo specific pH-dependent conformational changes that could modulate the interaction of PsbS with itself or other interaction partners in a membrane. We therefore consider our results consistent with *in vivo* studies that show that both M1 and M2 mutants have altered NPQ response (Li et al.) **We added a paragraph on p. 21, middle to describe this.**

As mentioned above, we add that it is very hard at this point of our understanding of the thylakoid membrane to establish a direct relation between a singular molecular membrane constituent and a specific *in vivo* response. We suggest that multiple interaction layers define the *in vivo* response that is triggered by the molecular switch.

5. A possible explanation for the discrepancies between the results in this manuscript and prior results is that the spectroscopic experiments described here were done on PsbS in detergent micelles. In previous experiments *in vivo*, PsbS was in the native thylakoid membrane of chloroplasts in intact leaves, and the experiments *in silico* were done on PsbS in a native-mimicking membrane. This important difference raises a question about the relevance of the results with detergent-solubilized PsbS protein to the situation in the thylakoid membrane *in vivo*, and this question should be acknowledged and discussed in this manuscript.

We essentially already addressed this valid point in our above responses. Below we summarize our opinion on this:

We discovered a pH-dependent motion of the H2 helix that must be an *intrinsic* property of the PsbS protein and that we suggest could be relevant for its function. Under which conditions the protein will show this pH-dependent movement, will likely be dependent on its microenvironment. In our case, the environment is a detergent micelle, in Liguori *et al.*, a (non-native) POPC bilayer was simulated and *in vivo*, proteins in thylakoid membranes are surrounded by galactosyl lipids that contain high amounts of the non-bilayer lipid MGDG. In addition, *in vivo*, PsbS could interact with partner proteins other than itself. We discussed the limitations of our approach and that of Liguori in one of the points above. As mentioned above, another aspect is the kinetics of the H2 movement, which may not have been observed in the MD study due to the limited time window of the simulations. As already mentioned, we do not necessarily see an inconsistency between the MD simulations by Liguori *et al* and our results.

As mentioned in our response to an earlier comment, we added a section on p. 19 to discuss the relation of our work with earlier MD simulations and a section on p. 21 where we address the relevance and limitations of our work with respect to the *in vivo* response.

6. An FTIR band at 1550 cm⁻¹ is described but not shown in Fig. 3. The x-axis range should be extended to lower wavenumbers to make this band evident.

We thank the reviewer for pointing this out. We were inaccurate in properly labeling this band: it should say 1570 cm⁻¹, which is in the range of the plot. We apologize for this oversight and have corrected it in the manuscript.

7. Why was pH 5.0 selected as the low pH condition? pH 5.0 is below the physiologically relevant pH range within which PsbS seems to work *in vivo*, according to experiments that estimated the pK_a of PsbS *in vivo* (Takizawa et al., 2007 BBA).

This value was taken because the PsbS X-ray structure was determined at this pH. As we performed a structural study on PsbS, we aim to include a condition where we have a direct comparison with the structural data that is available. **We made a remark on this on p. 5, middle.**

8. The introductory text could be edited to make it more concise and precise. Specifically:

- p. 2, last sentence of first paragraph needs references.

Following the advice of the reviewer we added references to the text.

- p. 2, second sentence of second paragraph should cite original paper rather than a review.

We added the reference to the original paper

- p. 2, fourth sentence of second paragraph cites a long list of papers, many of which do not seem directly relevant to the statement in this sentence. Papers showing the pH sensing function of PsbS, interactions with antenna proteins, and reorganization of PSII should be cited here. This sentence is largely repeated at start of third paragraph.

We changed the list of citations here, reducing the list to include those that are directly relevant according to the points raised by the reviewer. We also modified the sentence at the start of third paragraph to “PsbS activation involves two lumen-faced glutamate (Glu) residues and is associated with reversible monomerization of PsbS dimers” to make it clear that this paragraph focuses on the structure and function of the protein on molecular level and we changed the citations accordingly.

- p. 2, sixth sentence of second paragraph omits the fact that VDE and ZEP were overexpressed

in addition to PsbS in the cited paper (Ref. 20).

We added this fact to the sentence.

9. The Methods are lacking essential details that are necessary to allow the results to be replicated. For example, what concentration of detergent was used for refolding? What amount and concentration of protein sample was used for NMR and 2DIR spectroscopy?

Samples were prepared and refolded using 1% FC-12 detergent. For FTIR and 2DIR experiments, an Amide I absorbance of 0.8 at 1650 cm^{-1} was used in an optical path length of 20 micron, corresponding to a protein concentration of approximately 1 mM, at a volume of 100 μl .

This information was added to the Methods section.

Reviewer #4 (Remarks to the Author):

The present manuscript reports the infrared analysis of PsbS, a key protein in the photoprotection process of plant called non-photochemical quenching (NPQ). To clarify the molecular mechanism of the conformational pH switch of PsbS, the authors performed FTIR/2DIR measurements of the single and double mutants of two Glu residues (E71Q (M1), E176Q (M2), and E71Q/E176Q (M3)), which are the candidates of residues responsible for pH dependent conformational changes. NMR was also used but only to confirm pH-induced deprotonation/protonation of nonspecific carboxylate groups in the protein. They measured FTIR spectra of all the three mutants as well as the WT protein, but they focused on the analysis of amide I bands of WT and M2 to make a model of pH-dependent conformational changes of PsbS. 2DIR data were also obtained for WT and M2 to increase the resolution and provide the information of heterogeneity of bands. Although the proposed model is reasonable, assignments of the amide I peaks to the helical components (H2 and H3) and their conformational changes are rather speculative. In particular, the peak frequency of $1625\text{-}1630\text{ cm}^{-1}$ assigned to the H2 helical component is significantly low and usually in the β -sheet region, but could be possible in the case of some specific interaction of water as the authors suggested. However, without any conclusive evidence, the impression remains that the assignment of the IR bands are rather tentative.

We appreciate the reviewer's assessment that our proposed model is reasonable, yet as with a similar comment by Reviewer 3, we respectfully disagree with the notion that the assignment of the peak frequency of $1625\text{-}1630\text{ cm}^{-1}$ to H2 is tentative or speculative. 2D-IR indicates that this signal originates from a helical fragment, which excludes the possibility of a β -sheet or loop. The frequency is much too low to be assigned to a transmembrane helix and is only consistent with a water-exposed amphipatic helix fragment. Taking the known structure of PsbS (and in fact of all LHC-type protein complexes), the only amphipatic helix fragments are those at the lumen, i.e. H1 and H2. Moreover, we show that mutation of Glu-2, which is located in H2, abolishes the signal. The amplitude matches with a stretch of 8-9 amino acids, which closely agrees with H2

but not with H1. Hence, we confidently assign this signal to H2. **We modified the text to emphasize this on p. 14.**

The 1625 cm⁻¹ peak at pH 7.5 also diminished by M1 mutation near H3, which is at odds with the assignment to H2. This could be interpreted in the context of the interaction between H2 and H3 in a dimer or the reduced dimer population, but there is no explanation in the text.

The observation that in the absolute FTIR absorption spectrum of M1 (Fig. 3C, orange line), the 1630 cm⁻¹ shoulder in the M1 mutant is lower than that in WT (Fig. 3A, orange line) does not necessarily follow from a lower water-exposed amphipatic helix content (H2 or H1): it may well follow from a higher transmembrane helix content, which increases the absorption around 1655 cm⁻¹ (and hence results in a relatively lower 1630 cm⁻¹ shoulder). Note that in the X-ray structure, TM2 does not span the entire membrane at the luminal side and by mutation of nearby Glu-1, the unstructured part adjacent to TM2 has the potential to fold into TM2, especially because this stretch has a sequence that is very similar to that of the pseudosymmetry-related, fully transmembrane-spanning TM4. **We added a new section on p. 17-19 to discuss the non-native fold of the M1 and M3 mutants.**

IR measurements using mutants of other carboxylate residues such as E78 and E162 (*P. patens*) will be useful as control experiments to remove the possibility that the authors actually detected unspecific effects of mutations of carboxylate groups on the luminal side.

We do not understand the last comment of this reviewer that effects of other carboxylate residues would prove or disprove specificity. Mutation of other carboxylate residues such as E78 and E162 as suggested by this reviewer would not be appropriate as control experiment as it would target another specific residue and we do not have enough information on the functional and structural role of these other amino acid residues. Such mutational approach would be a new project in itself.

Other points:

1. The 2DIR map in Fig. 4A and B may be shifted to higher frequencies. The map in Fig. S5 is correct. All the 2DIR data of WT at pH 5 and the M2 mutant at pH 5 and 7.5 before taking subtraction should be presented at least in Supporting Information.

We thank the reviewer for carefully studying our spectra and noting the inconsistency in the frequency axis. As with a similar comment by Reviewer 2, we have corrected this oversight.

2. Fig. 6 is not based on the authors' data. It may be involved in the Fig. 1 as an introduction. The detail of the hydrogen bond interaction is not very clear without atomic color in the interacting residues shown in green and red.

In the manuscript and Fig. 6 caption we referred to the X-ray structure and the pdb entry, so it should be clear to the reader that this is not our data. **Nevertheless, we followed the advice of the reviewer and integrated Fig. 6 in Fig. 1. We modified the atomic color coding for clarity.**

3. E176(173) functions as a hydrogen bond acceptor in the dimer interaction. This interaction may be retained even in Q176 in M2. What is the reason for the decreased dimer population in this mutant? Also, mutation of E71(69) to Q may not interfere with the H2-H3 interaction at low pH, because it is not charged. Please explain the reason for the effect of these mutations on dimerization.

The hydrogen bond formed by E176 (E173) is via the backbone and could be formed by any amino acid type. However, the change of a carboxylate to an amide by changing Glu to Gln will change a carboxylate side chain to an amide, which could alter the orientation and/or induce local steric hindrance, and as such weaken the hydrogen-bond interactions that stabilize the dimer.

For intra-dimer interactions in M1 and M3, we cannot rely on the crystallographic structure as we learn from the IR data that the mutants have non-native folds. We could only speculate that a non-native fold of the first luminal loop by the M1 mutation destabilizes intradimer interactions. We do not have a clear explanation why M1 has a lower dimer content than M3, which also has a modified Glu-1 site. A possibility would be that the mutation of Glu-2 to Gln is detrimental for intra-dimer hydrogen bond formation to native Glu-1 sites, but beneficial for interactions with Glu-1 modified sites that have a non-native fold.

4. p. 8, last paragraph: No NMR data of M3 are shown in Fig. 2.

This was an error in the text that we have removed.

5. p. 20, line 10: Fig. 6 may be Fig. 4.

We thank the reviewer for spotting this mistake, which has been corrected.

References

- 1 Krishnan, M., Moolenaar, G. F., Gupta, K., Goosen, N. & Pandit, A. Large-scale in vitro production, refolding and dimerization of PsbS in different microenvironments. *Sci Rep* **7**, 15200, doi:10.1038/s41598-017-15068-3 (2017).
- 2 Krishnan, M., de Leeuw, T. & Pandit, A. Cell-free soluble expression of the membrane protein PsbS. *Protein Expression and Purification* **159**, 17-20, doi:10.1016/j.pep.2019.02.010 (2019).
- 3 Hamm, P. & Zanni, M. *Concepts and Methods of 2D Infrared Spectroscopy*. (Cambridge University Press, 2011).

- 4 Fenn, E. E. & Fayer, M. D. Extracting 2D IR frequency-frequency correlation functions from two component systems. *Journal of Chemical Physics* **135**, doi:10.1063/1.3625278 (2011).
- 5 Bernard, C. *et al.* The solution structure of a transient photoreceptor intermediate: Delta 25 photoactive yellow protein. *Structure* **13**, 953-962, doi:10.1016/j.str.2005.04.017 (2005).
- 6 Xie, A. H. *et al.* Formation of a new buried charge drives a large-amplitude protein quake in photoreceptor activation. *Biochemistry* **40**, 1510-1517, doi:10.1021/bi002449a (2001).
- 7 Brudler, R., Rammelsberg, R., Woo, T. T., Getzoff, E. D. & Gerwert, K. Structure of the I-1 early intermediate of photoactive yellow protein by FTIR spectroscopy. *Nat. Struct. Biol.* **8**, 265-270, doi:10.1038/85021 (2001).
- 8 Vreede, J., Juraszek, J. & Bolhuis, P. G. Predicting the reaction coordinates of millisecond light-induced conformational changes in photoactive yellow protein. *Proceedings of the National Academy of Sciences of the United States of America* **107**, 2397-2402, doi:10.1073/pnas.0908754107 (2010).
- 9 Harper, S. M., Neil, L. C. & Gardner, K. H. Structural basis of a phototropin light switch. *Science* **301**, 1541-1544, doi:10.1126/science.1086810 (2003).
- 10 Alexandre, M. T. A., van Grondelle, R., Hellingwerf, K. J. & Kennis, J. T. M. Conformational Heterogeneity and Propagation of Structural Changes in the LOV2/J alpha Domain from *Avena sativa* Phototropin 1 as Recorded by Temperature-Dependent FTIR Spectroscopy. *Biophysical Journal* **97**, 238-247, doi:10.1016/j.bpj.2009.03.047 (2009).
- 11 Zayner, J. P., Mathes, T., Sosnick, T. R. & Kennis, J. T. M. Helical Contributions Mediate Light-Activated Conformational Change in the LOV2 Domain of *Avena sativa* Phototropin 1. *Acs Omega* **4**, 1238-1243, doi:10.1021/acsomega.8b02872 (2019).
- 12 Konold, P. E. *et al.* Unfolding of the C-Terminal J alpha Helix in the LOV2 Photoreceptor Domain Observed by Time-Resolved Vibrational Spectroscopy. *Journal of Physical Chemistry Letters* **7**, 3472-3476, doi:10.1021/acs.jpcllett.6b01484 (2016).
- 13 Gil, A. A. *et al.* Femtosecond to Millisecond Dynamics of Light Induced Allostery in the *Avena sativa* LOV Domain. *Journal of Physical Chemistry B* **121**, 1010-1019, doi:10.1021/acs.jpcc.7b00088 (2017).
- 14 Peter, E., Dick, B. & Baeurle, S. A. Mechanism of signal transduction of the LOV2-J alpha photosensor from *Avena sativa*. *Nature communications* **1**, doi:10.1038/ncomms1121 (2010).
- 15 Freddolino, P. L., Gardner, K. H. & Schulten, K. Signaling mechanisms of LOV domains: new insights from molecular dynamics studies. *Photochemical & Photobiological Sciences* **12**, 1158-1170, doi:10.1039/c3pp25400c (2013).

REVIEWER COMMENTS

Reviewer #1 (Remarks to the Author):

The authors have addressed the itemized requests in the first reviewer report but did not provide a resolution of the problem addressed in the introductory statement:

„The conclusions based on the experimental results propose ne mechanisms for the action of PsbS in vivo. However, the interpretation still seems to be speculative in parts and more structural proof - in particular concerning the roles and locations of helices H2 and H3 with respect to the membrane would be desirable.“

I do not see a substantial improvement of the clarity in this respect in terms of structural interpretation. In particular the model presented in Fig. 5 is still somewhat speculative in my opinion.

Comments on the itemized responses:

(1) Did the authors consider adding an additional pH step between the 7.5 and 5.0 which could provide a cue to the transitional state between the two conformations? This could actually disambiguate the interpretation of Fig. 2, and one could probably reveal in which direction the two peaks are shifting, instead of assuming that they are buried beneath the other signals.

PsbS switches between two distinct states, so we do not expect any meaningful new information from choosing a pH value in between. The FTIR data fully agrees with the NMR data in showing protonation of all titratable residues at pH 5.0. Because the experimental steps for each sample type are very challenging and time consuming, i.e. production of mutants, refolding of the protein in detergent buffer, biochemical characterization, NMR spectroscopy, optimizing settings for FTIR, performance and analysis of the 2DIR experiments, we limited this study to two pH conditions.

+ The answer does not address the problem of ambiguities of assignment present in Figure 2.

(2) The author compare spectra at two pH values, but they use two different buffer systems (acetate and phosphate) - did they consider possible influence of the buffer anions on the stability of the two conformations?

We have to change the buffer system to stabilize the protein at pH 5.0 conditions. We do not expect large effect of the buffer anions on the conformation, considering that the proteins are surrounded by detergent micelles. In fact, varying the type of detergents does have some effect on the spectral shape of the CD spectrum of PsbS1 while the helical content remains the same and the effect is similar at neutral and low pH conditions. The spectral variation in the CD spectra for different detergent buffers was ascribed to changes in the coiled stromal loops of PsbS.

+ I agree, I'd expect the effect to be minor, but the reason for the buffer change should be mentioned in the paper.

(3) Were any multidimensional NMR experiments performed? For instance, 2D DARR could reveal how many amino acids are affected by structural changes.

Unfortunately, samples of PsbS in detergent micelles are not suitable systems for multidimensional solid-state NMR; samples at room temperature are liquid-like while the NMR resolution of frozen detergent solutions is poor. For 2D DARR experiments, we would have to prepare PsbS proteoliposomes. We started pilot NMR experiments on those, which need to be refined because of the low stability of proteoliposomes when they are loaded with high amounts of PsbS. Another direction to go for would be solution NMR, performing TROSY experiments on deuterated samples of PsbS in detergent micelles. We analyzed HSQC spectra of PsbS in different types of detergents to find the most suitable detergent conditions for solution NMR experiments¹ and may pursue in this direction.

+ Indeed it might take a considerable additional effort to provide good 2D solid-state NMR spectra or deuterated samples for solution NMR of these systems. Nevertheless this lack of more complete NMR data is a weak spot, makes results appear preliminary.

(4) In spite of the expected difficulties with resonance assignment of this protein, has at least a partial assignment been attempted?

A valuable support of the structural interpretation could be achieved by selective isotopic labelling (or unlabeled). This would allow one to really discriminate signals of the key amino acids.

A partial NMR assignment has not been performed yet for reasons mentioned above. Indeed,

selective isotope labeling or un-labeling will be a valuable support here. However, selective Glu labeling using host organisms is notoriously difficult due to label scrambling via metabolic pathways. To circumvent this problem, we already set up a cell-free expression system for production of PsbS. However, because of the technical challenges, selective isotope labeling and NMR assignments are considered outside the scope of this study.

+ See comment for item (3).

(5) Have you considered that the switch mechanism proposed might be temperature dependent. Could this be checked by different methods, e.g. solution NMR, CD?

Below we include a set of CD experiments recorded between 5 and 45 degrees Celsius collected of wildtype PsbS in FC-12 detergent micelles at pH 7.5 condition (Fig.1) and NMR HSQC spectra of PsbS at pH 5.0 and at pH 7.5, recorded at two temperatures (20 °C and 40 °C, Fig.2 and 3). For Fig. 1, we were not able to gather the original data from the CD setup to make a suitable figure due to Covid-19 lab access restrictions at this moment. We apologize therefore to show a photograph of a lab-book picture instead.

The CD and NMR spectra do not show any temperature-dependent conformational changes. The HSQC spectra at pH 5.0 have more resolved signals and a differential pattern compared to the spectrum at pH 7.5 at both temperatures. Several additional signals are resolved at the higher temperature, owing to increased protein dynamics at high T.

+ The additional CD and NMR data provided adequately resolve my concern.

(6) This reviewer is not an expert in 2DIR, however, in particular in view of the fact that 2DIR is not a very common approach, the interpretations of the 2DIR spectra in the paper would need some further clarification.

To address the reviewer's concern, we clarified some of the basic features of 2DIR spectroscopy in the revised text, on p. 10, middle, p. 11, bottom, p. 12, top, and added a citation to a comprehensive textbook by Hamman and Zanni. We emphasize that 2DIR spectroscopy has been applied to polypeptides and (small) proteins for two decades now, and it may be regarded as an established technique.

+ The additional clarification is helpful. Other reviewers of the paper, with a better background in 2DIR than myself, have commented in more detail on the 2DIR features.

(7) Minor point: Addition of a protein size "ladder" into the figures of SDS-PAGE gels would be beneficial.

We showed a protein-size ladder, but did not add the protein sizes to the ladder. We apologize for this and added those to Figure S1.

+ OK.

Reviewer #2 (Remarks to the Author):

The paper provides evidence for changes in helical secondary structure by FTIR/2DIR that help to illuminate the pH-induced response of PsbS in its role in photoprotection, currently a question of interest in the field of photosynthesis.

The assignments of three 2D IR features to changes in specific helical structures are overall reasonable; as convincing as possible with the data. I still have a couple lingering questions about details of the interpretation.

(1) The manuscript attributes the smaller difference signals in the M2 mutant to a small motion, similarly to the WT. However, no corresponding positive bands appear (Fig. 4C). The data seem to imply loss rather than shifting of helical structure.

(2) I am not convinced by the author's assignment of the [1630,1665] off diagonal band to loss of coupling by pH (Fig. 4B). If this were the case, another set of crossbands would be observed in the lower right quadrant. Perhaps laser heating is disrupting coupling? There are also rather substantial signals at 1670 cm⁻¹ on the excitation axis that do not make sense (Fig 4A).

Suggestions to add clarity:

- Bottom of pg 11: Paragraph arguing that 1625 cm⁻¹ signal due to helix rather than beta sheet. The added text arguing that the signal is not due to disorder adds confusion. In any case, providing a lower in addition to the upper limiting value for the anharmonicity is recommended.
- That Figure 4 shows FTIR spectra above 2D spectra is stated in text but not caption.
- Naming M3 rather as just M1/M2 would make easier to follow.
- Pg 9, states that the manuscript will concentrate on the M2 mutant, not M1 and M3. There was ~2 pages of discussion added about M1 and M3.
- The paper does seem it could be shortened with less repetition

Reviewer #3 (Remarks to the Author):

The authors have addressed most of my comments from the previous review, and I appreciate the added explanations and discussion in the text, which make it more understandable to a non-expert in IR spectroscopy. I certainly understand that one manuscript cannot answer everything about how PsbS responds to pH. I do however think that it is important to acknowledge and discuss apparent discrepancies between new results and previous work. Overall, this revised manuscript has been improved, and I believe it does provide experimental evidence for the long-hypothesized pH-dependent conformational changes in PsbS.

1. The new text (p. 19-20) comparing their results with previous MD results nicely discusses what appeared to be discrepancies between these studies.

2. The explanation of the "non-natural charge distribution" of the first loop in the M1 and M3 mutants (p. 18) makes sense for neutral pH, but not at pH 5 where the protonatable residues are all protonated. Thus, M1 and M3 at pH 5 would be expected to look like the wild-type protein at pH 5, but they do not. I do understand that the PsbS crystal structure showed a pseudo-C2 symmetry at pH 5, and it is unfortunate that there is not a crystal structure available for neutral pH. At this point, I do not wish to belabor this point, since the manuscript focuses on interpretation of M2.

3. The authors have tried to address my previous point about inconsistency between in vitro and in vivo results with new text on p. 21. One sentence here (lines 497-499) seems rather vague: "Our results indicate that it is difficult to establish a direct one-to-one relation between the singular molecular membrane constituent PsbS and the specific in vivo response, and suggest that multiple layers of interaction modulate its activity in vivo." Could the authors provide a more specific and detailed explanation similar to what was added on p. 19-20 (comparing their results with MD results)? It seems like some of the same reasons mentioned on p. 19-20 would apply to the comparison of PsbS in detergent micelles vs. PsbS in thylakoid membranes. I would suggest to move the two sentences on lines 457-462 to p. 21, because they are more relevant to the in vitro vs. in vivo situation.

4. Since PsbS is not a general "stress" sensor, I would suggest to add the word "light" to the title so that it reads "...the plant light stress sensor PsbS".

5. line 409: should be "D69" instead of "D68"

Reviewer #4 (Remarks to the Author):

The manuscript was properly revised in response to the comments. The reviewer thinks that the manuscript is now acceptable for publication.

Reviewer #1 (Remarks to the Author):

The authors have addressed the itemized requests in the first reviewer report but did not provide a resolution of the problem addressed in the introductory statement:

„The conclusions based on the experimental results propose ne mechanisms for the action of PsbS in vivo. However, the interpretation still seems to be speculative in parts and more structural proof - in particular concerning the roles and locations of helices H2 and H3 with respect to the membrane would be desirable.“

I do not see a substantial improvement of the clarity in this respect in terms of structural interpretation. In particular the model presented in Fig. 5 is still somewhat speculative in my opinion.

The reviewer does not further specify which part of Fig. 5 is speculative or what kind of structural proof is desired, which makes it difficult to address this point. The FTIR and 2DIR results are unambiguous, as we clearly explained in the response to all

reviewers' comments of the first revision. The main message of this manuscript is not to present a detailed (NMR) structure, but to report a pH-dependent conformational change, providing -for the first time- clear, experimental evidence that this protein acts as a pH switch. We are able to pinpoint the sites of action, namely H2 and H3 and we define the movement of H2 into a hydrophobic environment, i.e. the membrane. The discovery of the plasticity of the amphipathic helices might be a general motif, and further research is required to understand the consequences of this important finding for the mechanism of protein-protein interactions and in-vivo function of PsbS.

Comments on the itemized responses:

(1) Did the authors consider adding an additional pH step between the 7.5 and 5.0 which could provide a cue to the transitional state between the two conformations? This could actually disambiguate the interpretation of Fig. 2, and one could probably reveal in which direction the two peaks are shifting, instead of assuming that they are buried beneath the other signals.

PsbS switches between two distinct states, so we do not expect any meaningful new information from choosing a pH value in between. The FTIR data fully agrees with the NMR data in showing protonation of all titratable residues at pH 5.0. Because the experimental steps for each sample type are very challenging and time consuming, i.e. production of mutants, refolding of the protein in detergent buffer, biochemical characterization, NMR spectroscopy, optimizing settings for FTIR, performance and analysis of the 2DIR experiments, we limited this study to two pH conditions.

+ The answer does not address the problem of ambiguities of assignment present in Figure 2.

It is well known that protonation induces a upfield shift of NMR carboxyl resonances of Glu and Asp. The data shows that two clear peaks centered at 181.7 ppm and 178.0 ppm disappear at pH 5 while signal gain is observed at ± 179.3 and 177.0 ppm. Even though the NMR results may be ambiguous with regard to protonation states of Asp/Glu because the signals of protonated Asp/Glu are partially hidden under the carbonyl peak centered at 174.7 ppm, the FTIR results are clearly unambiguous: there, we see disappearance of COO⁻ at 1570 cm⁻¹, and appearance of COOH around 1700 cm⁻¹ upon lowering the pH. Our interpretation of the NMR results are fully consistent with this crystal-clear FTIR result. We adapted text on page 5, line 116 and down, to discuss the NMR spectral changes in more detail and refer to the FTIR results that unambiguously show a change in protonation for all protonable residues.

(2) The author compare spectra at two pH values, but they use two different buffer systems (acetate and phosphate) - did they consider possible influence of the buffer anions on the stability of the two conformations?

We have to change the buffer system to stabilize the protein at pH 5.0 conditions. We do not expect large effect of the buffer anions on the conformation, considering that the

proteins are surrounded by detergent micelles. In fact, varying the type of detergents does have some effect on the spectral shape of the CD spectrum of PsbS1 while the helical content remains the same and the effect is similar at neutral and low pH conditions. The spectral variation in the CD spectra for different detergent buffers was ascribed to changes in the coiled stromal loops of PsbS.

+ I agree, I'd expect the effect to be minor, but the reason for the buffer change should be mentioned in the paper.

We adapted the text, mentioning the reason for the buffer exchange, p. 5 line 98: "The two different buffers were chosen for optimal buffering capacity at the respective pH condition."

(3) Were any multidimensional NMR experiments performed? For instance, 2D DARR could reveal how many amino acids are affected by structural changes. Unfortunately, samples of PsbS in detergent micelles are not suitable systems for multidimensional solid-state NMR; samples at room temperature are liquid-like while the NMR resolution of frozen detergent solutions is poor. For 2D DARR experiments, we would have to prepare PsbS proteoliposomes. We started pilot NMR experiments on those, which need to be refined because of the low stability of proteoliposomes when they are loaded with high amounts of PsbS. Another direction to go for would be solution NMR, performing TROSY experiments on deuterated samples of PsbS in detergent micelles. We analyzed HSQC spectra of PsbS in different types of detergents to find the most suitable detergent conditions for solution NMR experiments¹ and may pursue in this direction.

+ Indeed it might take a considerable additional effort to provide good 2D solid-state NMR spectra or deuterated samples for solution NMR of these systems. Nevertheless this lack of more complete NMR data is a weak spot, makes results appear preliminary.

As mentioned above, we would like to make clear that the main message of this manuscript is not to present a detailed (NMR) structure, but to report a pH-dependent conformational change, providing -for the first time- experimental evidence that this protein acts as a pH switch. The IR results are neither ambiguous nor preliminary, and clearly pinpoint the sites of action and movement of the H2 helix.

(4) In spite of the expected difficulties with resonance assignment of this protein, has at least a partial assignment been attempted?

A valuable support of the structural interpretation could be achieved by selective isotopic labelling (or unlabeled). This would allow one to really discriminate signals of the key amino acids.

A partial NMR assignment has not been performed yet for reasons mentioned above. Indeed, selective isotope labeling or un-labeling will be a valuable support here.

However, selective Glu labeling using host organisms is notoriously difficult due to label scrambling via metabolic pathways. To circumvent this problem, we already set up a cell-free expression system for production of PsbS. However, because of the technical

challenges, selective isotope labeling and NMR assignments are considered outside the scope of this study.

+ See comment for item (3).

See our answer for item (3).

(5) Have you considered that the switch mechanism proposed might be temperature dependent. Could this be checked by different methods, e.g. solution NMR, CD? Below we include a set of CD experiments recorded between 5 and 45 degrees Celcius collected of wildtype PsbS in FC-12 detergent micelles at pH 7.5 condition (Fig.1) and NMR HSQC spectra of PsbS at pH 5.0 and at pH 7.5, recorded at two temperatures (20 oC and 40 oC, Fig.2 and 3). For Fig. 1, we were not able to gather the original data from the CD setup to make a suitable figure due to Covid-19 lab access restrictions at this moment. We apologize therefore to show a photograph of a lab-book picture instead. The CD and NMR spectra do not show any temperature-dependent conformational changes. The HSQC spectra at pH 5.0 have more resolved signals and a differential pattern compared to the spectrum at pH 7.5 at both temperatures. Several additional signals are resolved at the higher temperature, owing to increased protein dynamics at high T.

+ The additional CD and NMR data provided adequately resolve my concern.

We are happy to learn that these issues have been cleared up.

(6) This reviewer is not an expert in 2DIR, however, in particular in view of the fact that 2DIR is not a very common approach, the interpretations of the 2DIR spectra in the paper would need some further clarification.

To address the reviewer's concern, we clarified some of the basic features of 2DIR spectroscopy in the revised text, on p. 10, middle, p. 11, bottom, p. 12, top, and added a citation to a comprehensive textbook by Hammand Zanni. We emphasize that 2DIR spectroscopy has been applied to polypeptides and (small) proteins for two decades now, and it may be regarded as an established technique.

+ The additional clarification is helpful. Other reviewers of the paper, with a better background in 2DIR than myself, have commented in more detail on the 2DIR features.

We are happy to learn that the reviewer considers the clarification helpful.

(7) Minor point: Addition of a protein size "ladder" into the figures of SDS-PAGE gels would be beneficial.

We showed a protein-size ladder, but did not add the protein sizes to the ladder. We apologize for this and added those to Figure S1.

+ OK.

With this, we assume that items 5-7 were addressed satisfactorily in the first revised version of the manuscript.

Reviewer #2 (Remarks to the Author):

The paper provides evidence for changes in helical secondary structure by FTIR/2DIR that help to illuminate the pH-induced response of PsbS in its role in photoprotection, currently a question of interest in the field of photosynthesis.

The assignments of three 2D IR features to changes in specific helical structures are overall reasonable; as convincing as possible with the data. I still have a couple lingering questions about details of the interpretation.

(1) The manuscript attributes the smaller difference signals in the M2 mutant to a small motion, similarly to the WT. However, no corresponding positive bands appear (Fig. 4C). The data seem to imply loss rather than shifting of helical structure.

We agree with the reviewer and have adapted the text to indicate this on p. 17, line 375. Given that the signals are very small, we find it difficult to state with certainty that the signals represent unfolding rather than motion.

(2) I am not convinced by the author's assignment of the [1630,1665] off diagonal band to loss of coupling by pH (Fig. 4B). If this were the case, another set of crossbands would be observed in the lower right quadrant. Perhaps laser heating is disrupting coupling?

We should have formulated this more carefully by stating that a cross peak that exists in the equilibrium spectrum at neutral pH disappears in the equilibrium spectrum at low pH. The origins of cross peaks may be manifold including vibrational coupling, downhill vibrational population transfer etc. and it is certainly not the case that every cross peak in the upper left quadrant should invariably be accompanied by one in the lower right quadrant. Indeed, generally the amplitude of protein 2DIR maps is much larger and has more structure in the upper left quadrant than in the lower right quadrant, as is also observed here. Considering the above, we consider precisely assigning the origin of the [1630 1655] cross peak to be beyond the scope of this manuscript, but find it reasonable to attribute it to a change in vibrational coupling among elements involved in the observed conformational change. We reformulated the sentence on p. 14, line 300 to illustrate this.

We observed no laser heating effects during the 2DIR experimentation. Spectra derived from adjacent sample spots corresponded fully with those after prolonged illumination.

There are also rather substantial signals at 1670 cm⁻¹ on the excitation axis that do not make sense (Fig 4A).

It is not entirely clear to us to which signal the reviewer is exactly referring to. Steady-state protein vibrational spectra are immensely congested and contain several overlapping features that are difficult to assign *a priori*. This is exacerbated in 2DIR where contributions from through-bond/space vibrational coupling are manifest as off-diagonal features. Examples are given in the paper by Ganim *et al.* (ref 31), which demonstrate this complexity and we have also added a reference to a review article by Ghosh *et al.* (ref 32) as an additional example.

As such, the equilibrium spectrum was included merely to demonstrate consistency with analogous systems and we have no reason to suspect our results are atypical for a protein of this size and structural complexity. Rather, we focus on the difference spectra to decipher the pH-dependent response, given their more tractable nature.

Suggestions to add clarity:

-Bottom of pg 11: Paragraph arguing that 1625 cm^{-1} signal due to helix rather than beta sheet. The added text arguing that the signal is not due to disorder adds confusion. In any case, providing a lower in addition to the upper limiting value for the anharmonicity is recommended.

We agree that this was insufficiently described and have reworded this paragraph to better explain our rationale. More specifically, we refer to the anharmonicity and pattern of the 2DIR peaks to guide our assignment of the corresponding structural elements. In the specific case of anharmonicity, we provided reference to Wang and Hochstrasser, who calculated these values for a series of peptide configurations. Regarding limits, in the case of no coupling (upper limit) we can consider the monopeptide (NMA), which is known to be 16 cm^{-1} . Whereas, for peptide chains or ordered secondary structure, these values are highly dependent on the amino acid sequence and precise molecular geometry which modulates the extent of excitonic coupling. For example, alpha helix and beta sheet fall roughly in the range of 9-14 cm^{-1} . These values are well-established and this procedure is routine in the 2DIR community. We are simply using them to guide our structural assignment and feel that a deeper discussion about vibrational anharmonicity is not relevant in the context of our manuscript. We modified the text on p. 12, line 238.

-That Figure 4 shows FTIR spectra above 2D spectra is stated in text but not caption.

The thank the reviewer for pointing this out. Indeed this was an oversight, we have adapted the caption.

-Naming M3 rather as just M1/M2 would make easier to follow.

We followed the advice of the reviewer and replaced 'M3' by 'M1/M2' throughout.

-Pg 9, states that the manuscript will concentrate on the M2 mutant, not M1 and M3. There was ~2 pages of discussion added about M1 and M3.

We thank the reviewer for pointing this out and have deleted this remark.

-The paper does seem it could be shortened with less repetition

In response to the reviewer's comment, we deleted some text at the bottom of p. 9 regarding the fold of the M1 and M1/M2 mutants (which was described in the dedicated section on p.17/18 as well), and condensed the text on p. 17- 19 in the section '*E71Q (M1) and E71Q/E176Q (M1/M2) PsbS have non-native folds*'. We also streamlined some selected sentences for conciseness. We feel that we cannot delete or condense more text without a loss of clarity or completeness.

Reviewer #3 (Remarks to the Author):

The authors have addressed most of my comments from the previous review, and I appreciate the added explanations and discussion in the text, which make it more understandable to a non-expert in IR spectroscopy. I certainly understand that one manuscript cannot answer everything about how PsbS responds to pH. I do however think that it is important to acknowledge and discuss apparent discrepancies between new results and previous work. Overall, this revised manuscript has been improved, and I believe it does provide experimental evidence for the long-hypothesized pH-dependent conformational changes in PsbS.

1. The new text (p. 19-20) comparing their results with previous MD results nicely discusses what appeared to be discrepancies between these studies.

We thank the reviewer for this comment.

2. The explanation of the "non-natural charge distribution" of the first loop in the M1 and M3 mutants (p. 18) makes sense for neutral pH, but not at pH 5 where the protonatable residues are all protonated. Thus, M1 and M3 at pH 5 would be expected to look like the wild-type protein at pH 5, but they do not. I do understand that the PsbS crystal structure showed a pseudo-C2 symmetry at pH 5, and it is unfortunate that there is not a crystal structure available for neutral pH. At this point, I do not wish to belabor this point, since the manuscript focuses on interpretation of M2.

If the M1 and M3 point-mutations would only affect the charges, indeed M1 and M3 would be expected to look like the wild-type protein at pH 5. We suspect that replacing Glu by Gln has more effect than just modifying a local charge, which could be due to the presence of the Gln side chain NH₂.

3. The authors have tried to address my previous point about inconsistency between in

vitro and in vivo results with new text on p. 21. One sentence here (lines 497-499) seems rather vague: "Our results indicate that it is difficult to establish a direct one-to-one relation between the singular molecular membrane constituent PsbS and the specific in vivo response, and suggest that multiple layers of interaction modulate its activity in vivo." Could the authors provide a more specific and detailed explanation similar to what was added on p. 19-20 (comparing their results with MD results)?

On the reviewer's advice we deleted the sentence indicated above. We now give a more specific explanation for the apparent discrepancy between *in vivo* and *in vitro* results for the M2 mutant, which is based on our hypothesis that in the M2 mutant, H3 is locked in the low-pH conformation as a result of the specific self-interactions with PsbS in the homodimer. *In vivo*, PsbS likely interacts with other dimerization partners such as LHCII, which implies that H3 may be available for pH switching via Glu-1 and hence confer a (reduced) NPQ response.

We now write (p. 21, line 472): 'The observed partial activity *in vivo* may result from the function of PsbS to interact with other type of proteins, like LHCII. The M2 mutant has a fixed position of H2 but could still confer a pH-dependent switching capability through the protonation state of Glu-1 in H3, and establish reversible interactions with flexible amphipatic helices of other types of proteins. In our isolated protein system, however, PsbS can only interact with itself. This means that in the M2 PsbS dimer, the interacting partner site of H3 (i.e. the amphipatic helix H2 of the other monomer) is in a fixed position, thereby locking also the conformation and positioning of H3.'

It seems like some of the same reasons mentioned on p. 19-20 would apply to the comparison of PsbS in detergent micelles vs. PsbS in thylakoid membranes. I would suggest to move the two sentences on lines 457-462 to p. 21, because they are more relevant to the in vitro vs. in vivo situation.

On the reviewer's advice, we moved the sentences to p.21, line 483.

4. Since PsbS is not a general "stress" sensor, I would suggest to add the word "light" to the title so that it reads "...the plant light stress sensor PsbS".

We thank the reviewer for the suggestion and changed the title accordingly

5. line 409: should be "D69" instead of "D68"

We thank the reviewer for pointing us to this error, we changed D68 to D69.

Reviewer #4 (Remarks to the Author):

The manuscript was properly revised in response to the comments. The reviewer thinks

that the manuscript is now acceptable for publication.

We thank the reviewer for his/her positive advice.

REVIEWERS' COMMENTS

Reviewer #2 (Remarks to the Author):

I am satisfied that the 2D IR data support the main conclusion that alpha helical structure changes due to pH. I recommend publication.